## REVIEW ARTICLE

# Representing the function and sensitivity of coastal interfaces in Earth system models

Nicholas D. Ward [1,2✉], J. Patrick Megonigal [3], Ben Bond-Lamberty [4], Vanessa L. Bailey [5], David Butman [2,6], Elizabeth A. Canuel [7], Heida Diefenderfer [1,2], Neil K. Ganju [8], Miguel A. Goñi[9], Emily B. Graham [5], Charles S. Hopkinson[10], Tarang Khangaonkar[1], J. Adam Langley [11], Nate G. McDowell[5], Allison N. Myers-Pigg[1], Rebecca B. Neumann [6], Christopher L. Osburn [12], René M. Price [13], Joel Rowland[14], Aditi Sengupta [5], Marc Simard [15], Peter E. Thornton [16], Maria Tzortziou[17], Rodrigo Vargas [18], Pamela B. Weisenhorn [19] & Lisamarie Windham-Myers [20]

Between the land and ocean, diverse coastal ecosystems transform, store, and transport material. Across these interfaces, the dynamic exchange of energy and matter is driven by hydrological and hydrodynamic processes such as river and groundwater discharge, tides, waves, and storms. These dynamics regulate ecosystem functions and Earth's climate, yet global models lack representation of coastal processes and related feedbacks, impeding their predictions of coastal and global responses to change. Here, we assess existing coastal monitoring networks and regional models, existing challenges in these efforts, and recommend a path towards development of global models that more robustly reflect the coastal interface.

The coastal interface, where the land and ocean realms meet (e.g., estuaries, tidal wetlands, tidal rivers, continental shelves, and shorelines), is home to some of the most biologically and geochemically active and diverse systems on Earth[1]. Although this interface only represents a small fraction of the Earth's surface, it supports a large suite of ecosystem services,

[1] Coastal Sciences Division, Pacific Northwest National Laboratory, Sequim, WA 98382, USA. [2] College of the Environment, University of Washington, Seattle, WA 98105, USA. [3] Smithsonian Environmental Research Center, 647 Contees Wharf Road, Edgewater, MD 21037, USA. [4] Joint Global Change Research Institute, Pacific Northwest National Laboratory, College Park, MD 20740, USA. [5] Pacific Northwest National Laboratory, Richland, WA 99354, USA. [6] Civil & Environmental Engineering, University of Washington, Seattle, WA, USA. [7] Virginia Institute of Marine Science, William & Mary, P.O. Box 1346, Gloucester Point, VA 23062, USA. [8] Woods Hole Coastal and Marine Science Center, U.S. Geological Survey, Woods Hole, MA 02543, USA. [9] College of Earth, Ocean & Atmospheric Sciences, Oregon State University, Corvallis, OR 97331, USA. [10] Department of Marine Sciences, University of Georgia, Athens, GA 30602, USA. [11] Department of Biology, Villanova University, Villanova, PA 19085, USA. [12] Department of Marine, Earth, and Atmospheric Sciences, North Carolina State University, Raleigh, NC 27695, USA. [13] Department of Earth and Environment, Florida International University, Miami, FL 33199, USA. [14] Earth & Environmental Sciences Division, Los Alamos National Laboratory, Los Alamos, NM 87545, USA. [15] Jet Propulsion Laboratory, California Institute of Technology, Pasadena, CA 91109, USA. [16] Oak Ridge National Laboratory, Oak Ridge, TN 37830, USA. [17] Earth and Atmospheric Sciences, City University of New York, New York, NY 10003, USA. [18] Department of Plant and Soil Sciences, University of Delaware, Newark, DE, USA. [19] Argonne National Laboratory, Lemont, IL 60439, USA. [20] USGS Water Mission Area, Menlo Park, CA 94025, USA. ✉email: nicholas.ward@pnnl.gov

including sediment and carbon storage, contaminant removal, storm and flooding buffering, and fisheries production[2], with a global economic value of more than 25 trillion USD annually[3]. Roughly 40% of the world's population resides within 100 km of the coast[4]; much of the world's energy, national defense, and industrial infrastructure is located along coasts; and shipping of goods and resources, which depends on coastal ports, is responsible for ~90% of international trade[5]. By 2100, up to 630 million people will live on land below annual flood levels under high $CO_2$ emission scenarios, 2.5 times more than in the present day due to sea-level rise (which expands floodplains), immigration, and urban growth[6]. These close connections between the coastal interface and human societies represent a grand challenge for sustainably managing the resources that coastal ecosystems provide as urban development and human populations along the coasts continue to rise.

In addition to its importance for human livelihood, the coastal interface is an active component in the global cycling of carbon and nutrients. However, its global role remains poorly quantified in part due to the diversity of geomorphic settings, ecosystem types, their interconnectivity, and their dynamic behavior across a range of spatiotemporal scales[7–10]. Processes occurring in the water column and within sediments of tidal rivers, tidal wetlands, estuaries, and continental shelves significantly alter the quantity and quality of material that is both land- and marine-derived, and support the transfer of internally-produced materials across the coastal interface[11]. Further, a wide variety of coastal ecosystem types are demonstrated biogeochemical hotspots, in which process rates are not equivalent to the sum of terrestrial and aquatic contributions[12,13]. These highly dynamic biogeochemical processes are driven by two-way interactions between aquatic and terrestrial environments along the coast that remain poorly constrained empirically, resulting in limited representation in predictive models.

Global Earth system models (ESMs) used to predict how ecosystems interact to affect Earth's climate currently route riverine exports from land directly to the ocean with no processing within the coastal interface (Fig. 1). Inputs from land into the ocean are represented as fluxes that do not interact in the boundary/interface space. The lack of any form of processing that might alter either the quality or quantity of material transport between adjacent systems[14] may severely limit our ability to correctly depict the amount and form of water, energy, and matter entering the oceanic and atmospheric systems, as well as the effects of a wide range of disturbances and stressors with compounding effects such as sea-level rise, storm surge, and eutrophication on coastal ecosystems and infrastructure[15,16]. Local-to-regional scale models do exist for sub-elements of the coastal interface such as marsh and estuarine hydrodynamics, sediment budgets[17,18] and, more recently, photochemical and microbial processing of organic carbon[19]. Thus, there is potential for coupling specific components of these process-rich fine-scale models with global scale ESMs to more accurately depict the coastal interface.

We review what is known about the ecological and biogeochemical functions of coastal ecosystems in the context of the attributes and processes that should be represented in ESMs. We then provide recommended approaches for advancing the representation of the coastal interface in ESMs in order to improve climate predictions and impacts on the world's economically valuable and densely populated coastal zone. We advocate for an improved mechanistic understanding of coastal interfaces from ecological and functional perspectives, the impact of coastal interfaces on global biogeochemical cycling and climate, and the effect of disturbances on coastal interfaces across a range of spatiotemporal scales.

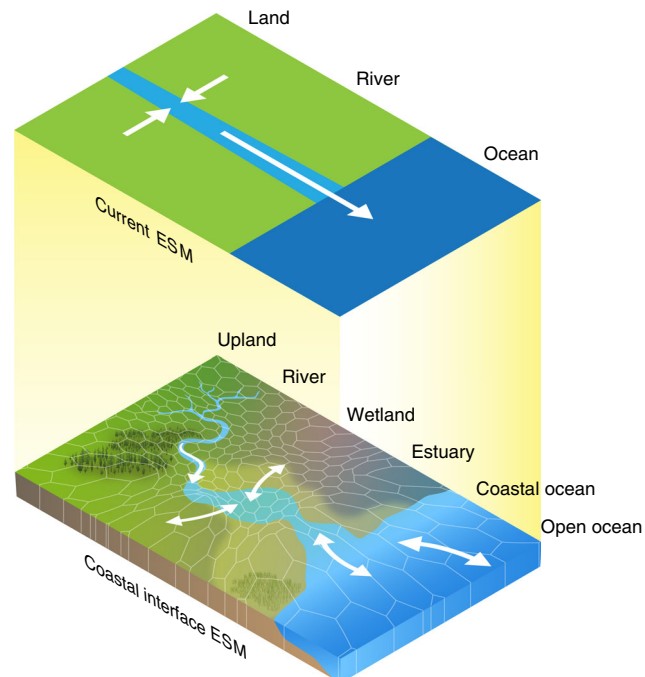

**Fig. 1 Earth system model representation of the coastal interface.** Current Earth system models (ESMs) represent the land and ocean as disconnected systems, with freshwater discharge being the only meaningful connection. Next-generation models should represent land–sea connections by incorporating coastal features such as the tidal rivers, wetlands, estuaries, the continental shelf, and tidal exchange across the coastal terrestrial–aquatic interface. This likely necessitates coupling different models to produce details at the sub-grid scale.

## Overview of coastal interfaces

**Ecosystem-scale interactions.** This section describes the fundamental ecosystem-scale attributes and interactions that define the coastal interface and should be represented in coupled land–ocean models. Coastal interfaces are transition zones between land and ocean where the magnitude, timing, and spatial pattern of freshwater–seawater mixing determine the nature of biogeochemical gradients (Fig. 2). The primary defining feature of a coastal interface is a sea-to-land gradient in tidal influence on surface water elevation[20]. Hydraulic head gradients may drive the majority of groundwater fluxes and exchange[21], but groundwater also responds to tidal variation, with tidal fluctuations driving a two-way exchange of water and geochemical constituents such as $CO_2$ and salt between the land, groundwater, and surface waters[22]. As such, we broadly define the coastal interface as any region where land, freshwater, and tides interact, or in other words all land surfaces (e.g., wetlands, marshes, floodplains) and water bodies (e.g., tidal rivers, estuaries, lagoons, deltas, and continental shelves) lying between purely inland and marine settings. These settings are complex and diverse by definition (Fig. 2) and encompass watersheds that lie below the head of tides.

Interactions between fresh groundwater discharge, river discharge, estuarine circulation, and tidal elevation determine the position and length of another defining feature of the majority of coastal interfaces — salinity gradients[23]. In the case of the tidally-influenced reaches of rivers with high discharge such as the Amazon River, the landward salinity intrusion is limited and water can remain fresh some distance offshore onto the continental shelf[10]. In contrast, smaller rivers experience significant salinity intrusion into river channels, groundwater, and soils[24]. The extent of the salinity gradient directly influences terrestrial vegetation

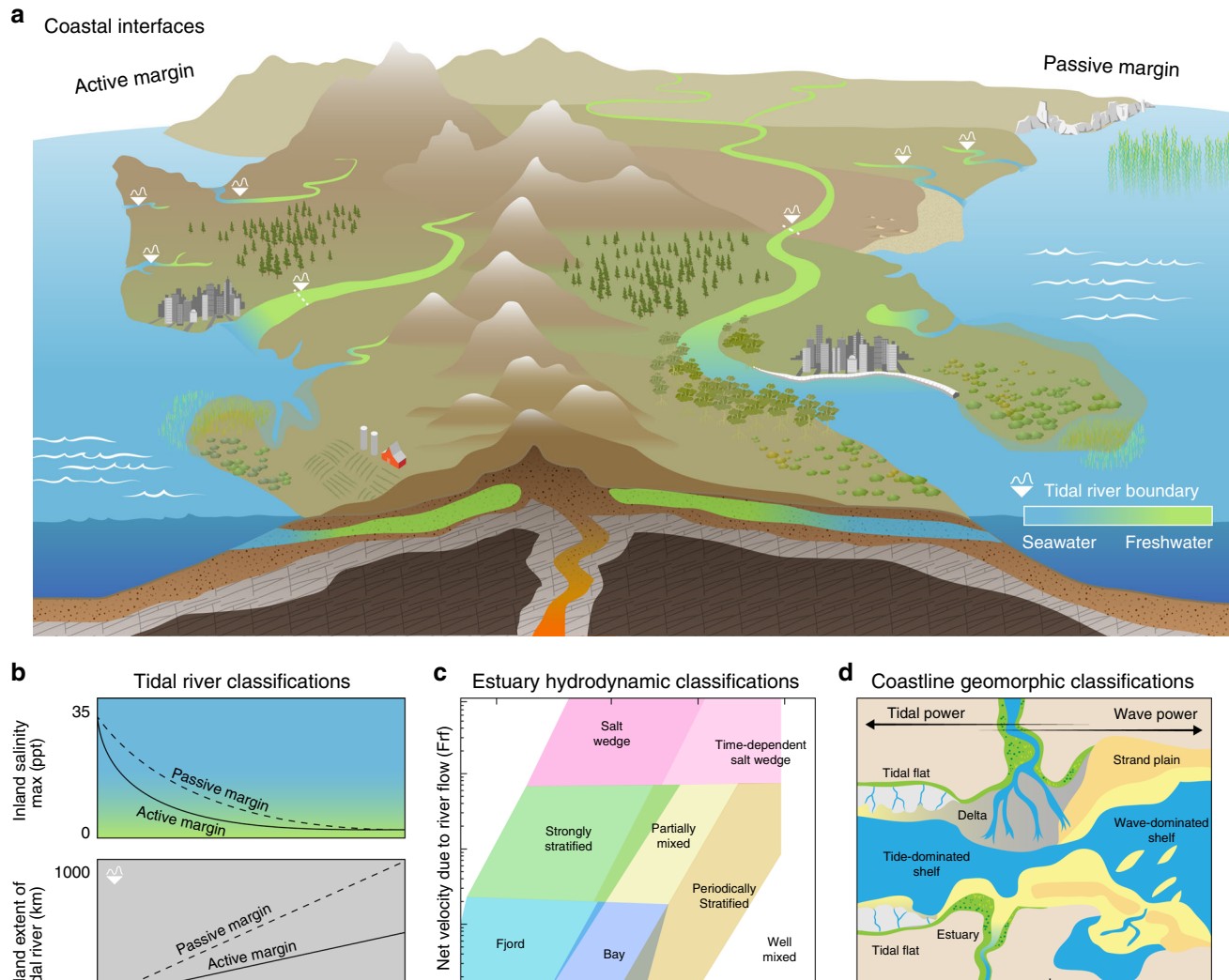

**Fig. 2 Generalizable features of coastal ecosystems. a, b** The inland extent of tidal influence on river flow increases with stream order, while the inland intrusion of salinity decreases. Rivers (and groundwater tables) on an active continental margin (e.g., US West Coast) are generally steeper in elevation, reducing how far inland tides permeate. Gradients in vegetation are influenced by these characteristics. **c** Estuarine environments can be broadly classified by their hydrodynamic properties such as net current velocity due to river flow ($Fr_f$) and how effectively tides mix a stratified estuary (M); fjords have low freshwater and tidal velocity scales due to their great depth whereas salt wedges have high contributions from rivers and a wide range of tidal contributions (adapted from Geyer and MacCready[23]). **d** Classifications of shallow water depositional environments along the coast can be categorized based on the ratio of wave power to tidal power and whether they are regressive (i.e., net land gain; top half of the diagram) or transgressive (i.e., net land loss; bottom half of the diagram) environments. The top half of the diagram shows regressive environments such as deltas and strand plains. The bottom of the diagram shows transgressive environments such as estuaries and barrier lagoons. Open coast tidal flats and shelf environments can be linked to either type of coast with shelf width decreasing during regression (adapted from Steel and Milliken[107]).

distribution along the land-to-sea hydrologic gradient, as well as soil and sediment biogeochemistry and geomorphology in a bi-directional manner. For example, tidal exchange can both deposit marine-derived material onto terrestrial landscapes[25] and export terrigenous material to the sea[26,27]. Tidal influences on coastal ecosystems go beyond effects on salinity distributions to include effects on water velocity, flow direction, and flood frequency with consequences for carbon and nutrient exchanges in tidally affected freshwater wetlands[27,28].

The critical functions of shoreline stabilization and nutrient, carbon, and water cycling rely on vegetation within the coastal interface[15]. The distribution and productivity of coastal interface vegetation (e.g., algae, succulents, grasses, sedges, rushes, forbs, woody shrubs, and trees) is driven by gradients in flooding, salt and sulfide exposure, nutrient availability, topography, herbivore activity, and soil characteristics such as $O_2$ availability and redox potential[29]. Plant species diversity generally decreases with increasing salinity and flooding intensity, shifting from ecosystems that have many similarities with upland settings where tidal influence is minimal, to low-diversity communities dominated by halotolerant species such as cordgrass, mangroves, or succulents, and finally ending with perennially submerged aquatic plants such as seagrass[30].

Submerged vascular plants and emergent marshes are at the front line of the coastal interface because changes in their extent can have broad impacts across the whole coastal domain, and perhaps beyond[31]. Functional redundancy in the form of different species that contribute similarly to an ecosystem function is

typically thought to be relatively low in such diversity-poor systems where few species can tolerate the harsh and fluctuating conditions, similar to terrestrial diversity-poor grasslands[32]. However, not all functional diversity occurs across species; the monospecific stands that dominate vast coastal wetlands often exhibit great genotypic diversity, which may yield high functional diversity despite low species richness[33]. As wetlands adapt to climate trends, the potential changes in relative representation of plant functional types — how models simplify plant diversity into manageable categories — must be incorporated into predictions of future coastal ecosystem function and adaptation. Understanding and characterizing such responses are critical to accurate representation of plant functionality in coastal interface models.

**Biogeochemical interactions and cycles.** This section describes the fundamental biogeochemical functions of coastal ecosystems that are likely the most critical to represent in regional and global scale models. Interactions among hydrology, vegetation, geomorphology, soils, and sediments influence the quantity and composition of carbon, nutrients, and redox-active compounds (e.g., $O_2$, $SO_4^{2-}$) within and exchanged by coastal interface ecosystems. Furthermore, many of these processes may be interactive across spatial scales[34]. For this reason, one of the largest challenges in constraining the role of coastal interfaces in global biogeochemical cycles is scaling our quantitative understanding of biotic and abiotic controls on molecular transformations and fluxes gained at the pore (e.g., $nm^3$ to $\mu m^3$), core ($cm^3$), or plot ($m^2$ or $m^3$) scale to estuarine sub-basins ($km^2$), entire estuaries (10–1000 $km^2$), and ultimately to the scale and process resolution of ESMs (100-10,000 $km^2$).

The role of coastal ecosystems in the carbon cycle is important both for constraining global carbon budgets and also representing these significant fluxes in ESMs. Inland waterways concentrate material inputs from an entire watershed, which then pass through the coastal interface. In the case of organic carbon (OC), the small amount of OC that is mobilized from upland soils to rivers on an area basis (~1–5 g OC $m^{-2}$ $yr^{-1}$ globally) translates to 2 orders of magnitude greater loading (~300 g OC $m^{-2}$ $yr^{-1}$) into coastal interface ecosystems, which are a relatively small focal area (i.e., bottleneck) for inputs coming from the entire watershed[7]. It is currently estimated (via mass balance) that ~5.7 Pg of inorganic and organic C $yr^{-1}$ is mobilized from upland terrestrial systems through inland waters and wetlands, of which 74% is returned to the atmosphere as $CO_2$ prior to delivery to the coastal ocean; this total flux is of similar magnitude as anthropogenic $CO_2$ emissions from fossil fuel burning (7.9 ± 0.5 Pg C $yr^{-1}$), uptake by the ocean (2.4 ± 0.6 Pg C $yr^{-1}$) and terrestrial biosphere (2.7 ± 1.2 Pg C $yr^{-1}$)[10]. $CO_2$ emissions from tidal rivers have not yet been adequately included in global carbon budgets, but may make a substantial contribution considering the increasing surface area associated with the lower reaches of rivers[35].

Despite their relatively small global surface area (0.07–0.22%), vegetated coastal systems (seagrass, mangroves, and intertidal marshes) sequester 65–215 Tg C $yr^{-1}$, globally, which is equivalent to ~10% of the net residual land sink and 50% of carbon burial in marine sediments[36]. These ecosystems are being lost at a rate of 1–7% $yr^{-1}$ due to human activities such as dredging, filling, eutrophication, and timber harvest[37]. Such habitat losses may also stimulate OC export and decomposition to $CO_2$ in coastal interface ecosystems[38]. Continental shelves play a similarly active role in global carbon cycling due largely to an abundance of nutrients from upwelling. Although continental shelves represent 7–10% of global ocean area, they contribute to 10–30% of global marine primary productivity; 30–50% and 80% of global inorganic and organic carbon burial in marine sediments, respectively; and up to 50% of the deep ocean OC pool[7].

The extent of OC transformation or loss as it passes through the coastal interface depends on its transport time, path, and exposure to the variety of surfaces (e.g., suspended particles, soil pores, and sediments) within the interface[10]. While allochthonous inputs can influence coastal ecosystem function, local sources of production also export or filter allochthonous transport. For example, processes such as low-tide rainfall can result in elevated mobilization of particulate OC (POC) from intertidal landscapes that can represent a significant fraction of annual POC export in many of these environments[39]. In addition, direct leaching from marsh plants and litter, exudation from roots, and biological production by algae are major local sources of chemically and optically distinctive dissolved organic matter to estuaries and coastal oceans[9,27].

Gradients in microbial community composition from rivers to continental shelves are generally controlled by salinity and redox (spatially) and river discharge (temporally) with distinct assemblages present in tropical, temperate, and high-latitude settings[40,41]. The hydrologic and geochemical gradients that characterize soils of coastal landscapes, particularly salinity and dynamic redox conditions, exert a strong influence over soil microbial community composition and metabolic functioning[25]. It remains a challenge to differentiate the effects of inundation and water chemistry on microbially driven biogeochemical functions in soils. At the pore-scale microbial activity, hydrologic connectivity, and drought legacy interact to regulate ecosystem functions[42]. At the core and plot scale, salinity dominates the controls on soil organic carbon (SOC), and salinity is negatively correlated with SOC content[43]. Along natural gradients, increasing salinity is correlated with an increase in denitrification[44] and a decrease in methane emissions[45], while increases in salinity tend to decrease both methanotrophy and methanogenesis in previously freshwater environments[46]. Variability in the duration of salinity exposure can influence the production of greenhouse gases. For example, long-term soil exposure to seawater decreases microbial $CO_2$ production[47] while short pulses of seawater exposure increase $CO_2$ emissions[48]. However, rapid changes in salinity gradients could result in unexpected patterns of greenhouse gas emissions at sub-daily scales[46]. Other coupled microbial cycles may be less sensitive to salinity. For example, a diverse community of sulfate-reducing bacteria associated with tidal freshwater systems has been shown to be relatively resistant to seawater intrusion[49]. However, the full range of time frames (from seconds to years) over which these sensitivities could emerge have not been examined.

## Challenges for constraining coastal dynamics

**Hot spots and hot moments.** Because of their position at the interface of land and water, and thus constant exposure to terrestrial and aquatic fluxes, coastal ecosystems represent hot spots for processing and transformation of energy and matter (Fig. 3). Hot spots are defined as areas that show disproportionately high metabolic rates or carbon stocks relative to the surrounding areas[13], and to their spatial representation. We suggest that hot spots can range from fine scales (e.g., $cm^3$, $m^2$) to the scale of entire estuaries (10–1000 $km^2$) and influence local to global scale material budgets depending on the process.

It is both feasible and desirable to represent hot spot dynamics in ESMs that play a significant role on global scale biogeochemical cycles and are empirically understood. For example, mangroves cover 0.1% of the Earth's surface[50] but are among the most productive carbon-sequestering ecosystems on Earth

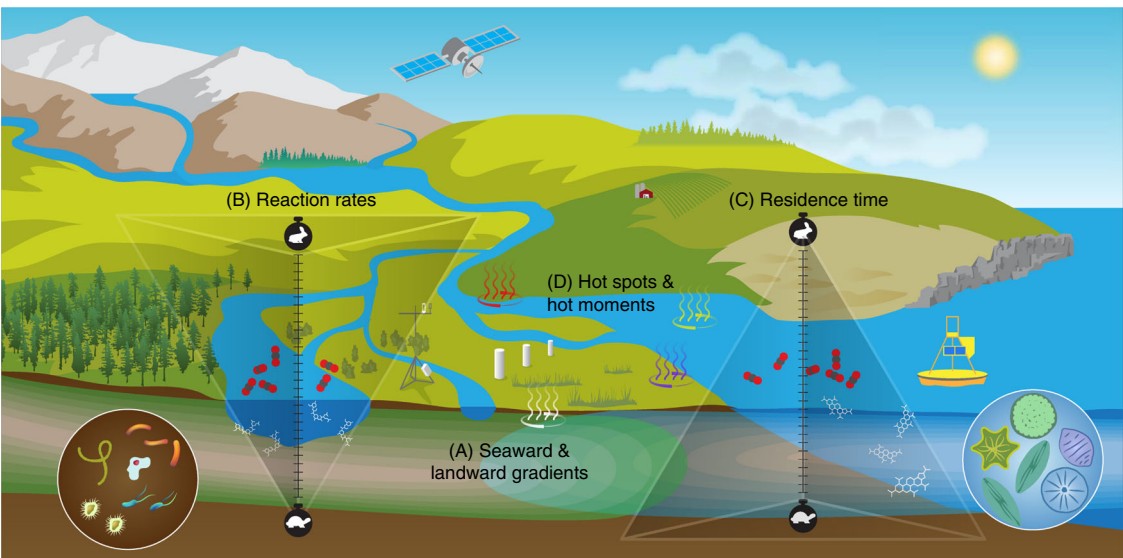

**Fig. 3 Biogeochemical characteristics of coastal interfaces. a** Two-way exchange of water and materials between terrestrial and marine environments drive gradients in geochemical constituents (e.g., ions, carbon, nutrients), plant distribution, and ecosystem functions (e.g., carbon storage, greenhouse gas emissions, sediment accumulation). **b** Biogeochemical reaction rates generally occur at more rapid timescales (e.g., hours to days) in aquatic systems such as rivers compared to soils and sediments (years to millennia). **c** Likewise, the residence time of biogeochemical components is short in aquatic environments such as estuaries and the surface ocean compared to the deep ocean and its sediments. **d** Coastal interface biogeochemistry is complicated by an abundance of hot spots and moments for diverse reactions across scales that can significantly alter expected reaction rates and residence times.

($1023 \, \text{Mg C ha}^{-1}$) and thus are hot spots for carbon storage and uptake from regional[51] to global scales[52]. More broadly, estuaries could be considered hot spots for productivity, carbon storage[53], and/or decomposition[54] depending on hydrologic factors such as water residence time, estuarine exchange flow patterns, and position of the estuarine turbidity maximum zone[11]. For example, $\sim 18 \, \text{Tg C yr}^{-1}$ is buried in fjord sediments, globally, which is equivalent to 11% of marine carbon burial rates; much of this OC is terrestrially-derived owing to the steep topography and a short residence time between terrestrial soils and estuarine sediments in these environments[53]. This is a feature of landscapes on active margins, whereas lower relief landscapes on passive margins have longer residence times and a greater extent of OC transformation prior to burial[55] (Fig. 2). These examples of depocenters (areas of maximum deposition) for rapid carbon burial are not only relevant to modern-day carbon cycling, but also act as significant carbon sinks over geologic timescales. For example, sustained burial of woody debris in Bengal Fan sediments has occurred over the last 19 million years; this debris is largely of lowland origin[56], suggesting that alterations to the land use and hydrology of coastal interface ecosystems could influence geologically-relevant processes over modern timescales.

Coastal ecosystems are sensitive to rapid and disproportionate hydrological and biogeochemical fluctuations with terrestrial, atmospheric, and oceanic origins including extreme precipitation events[57], snow/ice melt[10], accumulation and enhanced dry deposition of atmospheric pollutants[58], extreme high tides, and storm surges[46,59]. Thus, hot moments — short time periods with disproportionately high metabolic rates—may play a prominent, but typically ignored role in coastal ecosystem biogeochemical cycling. These hot moments may be controlled by processes occurring around the roots of plants (i.e., the rhizosphere) driven by interactions between plants and microorganisms, plant-driven water flow and solute transport, plant uptake of nutrients, soil chemical reactions such as rapid changes in redox potential[60] or sorption and cation exchange[61], or mixing of terrestrial and aquatic-derived substrates[38]. Hot moments play a larger role in certain biogeochemical cycles than others. For example, although

soil methane emissions generally decrease, and even become negative (i.e., uptake from the atmosphere) along coastal salinity gradients, rapid events such as ebullition induced by storm surge can result in momentarily high $CH_4$ fluxes[59]. Similarly, periods of intense rainfall during low-tide conditions can result in elevated rates of erosion and transport of sediment and organic matter from intertidal platforms (e.g., vegetated marshes and unvegetated mudflats) to adjacent creeks and surrounding coastal ocean[62].

A key challenge of measuring and modeling coastal interfaces is determining the spatiotemporal scale(s) needed to represent processes and systems such that the outcomes of interest are not biased by misrepresentation of available measurements in time and space, relative to the hot spots and hot moments that characterize the system. For example, inter-comparisons of methane models show large inconsistencies that are primarily due to uncertainties in temperature sensitivity, substrate limitation of $CH_4$ production, and wetland area dynamics[63]. While the last issue can be addressed by using consistent surface water inundation remote sensing products[64], the first two issues represent knowledge and modeling gaps that exist, in part, because of the highly dynamic nature of methane production and emission. High temporal resolution measurements of different processes are thus needed to couple ecosystem responses (e.g., greenhouse emissions) with the underlying controls to properly represent hot moments in regional models and ESMs. While new technologies are emerging that allow highly resolved organic carbon or gas flux measurements[46,59], there is a lack of consensus on how to appropriately scale lateral land-water carbon fluxes, or carbon emissions from either the bottom-up or top-down origin[8,65].

The concept of hot spots and hot moments has been criticized for lacking a quantitative definition. For example, it has been suggested that unusually high spatiotemporal variability with ecosystem-scale importance should be defined as ecosystem control points with four distinct categories: permanent control points that experience sustained high rates of biogeochemical activity relative to surrounding areas such as riparian and

hyporheic zones, activated control points that only support high rates when a limiting resource such as nutrients or oxygen is delivered, export control points that accumulate reactants until some threshold is reached that allows export such as OC accumulation in soils that is mobilized only during storms, and transport control points that have a high capacity for transporting solutes/reactants such as macropore flow paths in soils or stormwater drainage pipes[66]. Capturing the spatial and temporal variability of ecological processes across coastal interfaces in this context remains unclear; and consequently represents a challenge to be included in ESMs.

**Disturbances and stress at the coastal interface.** Coastal ecosystems are broadly sensitive to disturbances and stress from surrounding watersheds and the ocean that result in anomalous (i.e., non-steady state) responses. Disturbance typically refers to events that temporarily alter ecosystem attributes (e.g., plant productivity, GHG fluxes, etc.) but occur infrequently enough to allow for recovery time during which attributes re-establish a normal dynamic equilibrium; in contrast, higher frequency or continuous stress events permanently shift the trajectory of an ecosystem attribute[67]. Long-term stress to an ecosystem is also referred to as a press as opposed to a pulse disturbance event, and these can interact producing compounding effects[68]. The dominant chronic stressors on coastal ecosystems are sea-level rise[15,16], temperature increases[69], ocean acidification[70] (Fig. 4), land use conversion (e.g., urbanization), and long-term alterations to water flow (e.g., river impoundment and water extraction) and coastal-estuarine circulation[71] (Fig. 4). The dominant

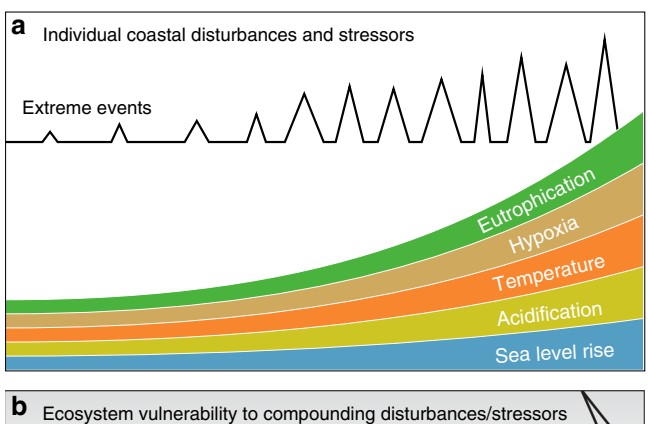

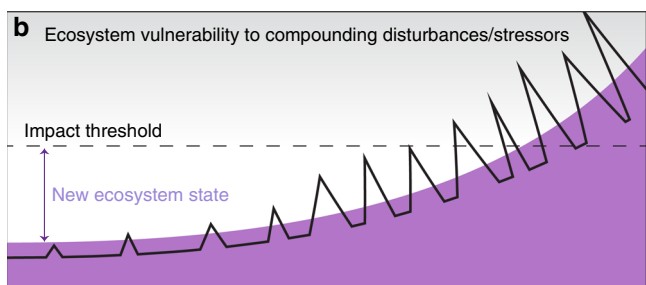

**Fig. 4 Coastal ecosystem disturbances, stressors, and vulnerability.**
**a** Increasing air and water temperatures, water acidification, rates of sea level rise, eutrophication, hypoxia, and frequency/magnitude of extreme storm surge events are among the primary threats to the ecology and hydro-biogeochemistry of coastal interfaces. **b** Although the resilience of coastal ecosystems is relatively unknown, it is likely that compounding disturbances and chronic stress will eventually exceed their impact threshold, resulting in widespread collapse of ecological function. Additional drivers of change not shown include land use change, river impoundment, natural resource extraction, invasive species, droughts, floods, and fires (concept inspired by McDowell et al.[89]).

episodic disturbances are flooding (either from storm surges or upland sources); drought; and temporary vegetation removal via sedimentation, erosion, wildfire, harvest, and other human manipulations.

Changes in the global distribution of ecosystems along the coastal interface must be considered in light of centuries of direct human alterations. Climate change will likely increase the frequency of extreme weather events (droughts to tropical cyclones), dramatically altering the delivery of water, nutrients, and carbon to coastal zones. Chemical constituents associated with extreme weather result in extended periods of degraded water quality as well as switching modes of coastal ecosystems between autotrophy to heterotrophy[72]. The timing and longevity of these perturbations add to the uncertainty of the role of these systems as greenhouse gas sources or sinks and as exporters of carbon to the oceans[7]. Further, the ecological structure of coastal ecosystems is already experiencing the effects of sea-level rise with coastal forest boundaries retreating inland[15] and salinization of tidal freshwater systems shifting their function and related rates of carbon burial and greenhouse gas (i.e., $CO_2$ and $CH_4$) emissions[48]. Tidal marshes have been reclaimed for agricultural purposes throughout Western Europe and North America, and large-scale reclamation and land conversion continues in regions including coastal China, impacting hydrologic connectivity and ecosystem-scale fluxes with the construction of various engineered seawalls[73]. Eutrophication of estuarine waters occurs as a result of both natural episodic nutrient inputs and long-term changes in land use practices (e.g., agriculture, septic systems, nitrogen-fixing vegetation, etc.), and in some cases results in hypoxic conditions that can harm fish and wildlife; hypoxia occurs due to both natural and anthropogenic causes[74]. Deforestation can alter the function and resilience of coastal ecosystems, ultimately causing an irreversible loss of coastal wetlands. Mangrove forests in tropical regions are losing between 0.16 and 0.39% of land area annually to development, aquaculture, and agriculture[75]. Such coastal land alteration has already released large quantities of soil organic carbon to the atmosphere as $CO_2$, and an estimated 0.15–1.02 Pg C yr$^{-1}$ continues to be emitted globally[76].

Shifts in the interaction between freshwater hydrology and tidal influences due to sea-level rise, delta subsidence, or anthropogenic changes (e.g., impoundments) will impact coastal interface geomorphology, such as delta evolution, riverine and coastal sedimentation, and wetland ecological/physical structure[20]. Changes in sediment supply may be considered a stress that alters the evolution of wetland structure and function, although episodic events such as landslides and volcanoes are disturbances under the return-interval-based terminology adopted here[67]. Though meta-analyses have shown that salt marshes can keep pace vertically with sea-level rise[77], the lateral position of marshes is not as stable as they narrow or expand depending on the net sediment budget[78] and external stressors and disturbances such as waves, storm surge, and sea-level rise. The contraction of marsh area likely produces an increased export of organic and inorganic material across the coastal interface[26,79] although some portion of the material is re-deposited on the marsh plain during the landward transgression process[80].

Although wave-induced erosion may be considered an episodic disturbance, moderate storms and diurnal winds are mainly responsible for the majority of salt marsh edge erosion[81]. Internal deterioration of salt marshes, through salinity intrusion, herbivory, eutrophication, or other chronic factors has also been linked with sediment export[82]. Both lateral erosion and internal deterioration can be considered as net neutral processes from a budgetary perspective if landward migration corridors are

available[26,83]. However, given the rapid nature of salt marsh loss and extensive coastal development, it is likely that salt marsh loss is a net contributor of material across the coastal interface.

Another visibly prominent shift in coastal ecology is the poleward migration of mangroves due to declining freeze frequencies and landward migration due to seawater intrusion[84]. Conversion from herbaceous-dominated to woody plant-dominated wetlands greatly increases aboveground carbon uptake on the landscape scale[52,85] and can accelerate soil elevation gain[86] and long-term wood retention in channels and floodplain microtopography[85], influencing long-term persistence of these ecosystems. Conversely, seawater intrusion into freshwater wetlands at the upstream edge of the coastal interface can cause vegetation death and accelerated soil carbon loss resulting in the collapse of the ground surface and a conversion of the plant-dominated wetland to open brackish water[87]. Such landscape-level shifts are dependent on a complex interplay between land use (e.g., extent of coastal development), geomorphic conditions, and relative sea level rise. For example, direct salt marsh conversion to open water or tidal flats may have a greater importance than mangrove expansion into salt marsh habitats in low relief areas with high relative sea level rise, while tidal freshwater marshes may either increase or decrease in areal extent under mean or max sea-level rise scenarios[88].

Among the largest uncertainties in projecting the future distribution, structure and function of coastal interfaces is quantifying tipping points for the collapse of ecosystem structure and function[89] across the coastal domain that incorporates the combined effects of a myriad of disturbances and stressors with compounding impacts (Fig. 4). Extreme events can push ecosystems already under stress beyond their tipping point, altering long-term ecosystem structure and also act as a hot moment for biogeochemical activity in the shorter term[72]. A fundamental goal of ESMs is the ability to accurately predict the influence of ecosystem distributions, structure, and function on global climate. Achieving this goal necessitates representation of feedbacks between the complex processes, stressors, and disturbances described above.

**Modeling the coastal interface.** Current generation ESMs such as the Energy Exascale Earth System Model[90] are coupled climate models that aim to simulate the Earth's climate system, which depends on terrestrial and ocean biogeochemistry, and the interactions between atmosphere, ocean, and land (as well as ice, in high latitude regions). Coastal interfaces fall in between these traditional ESM domains, and are not typically represented in such models[1]. The dynamic nature and non-linear, unpredictable, and heterogeneous biogeochemistry of coastal ecosystems present huge challenges for model representation. Further, the omission of coastal interfaces emphasizes a critical question: what coastal processes need to be considered, at what spatial and temporal scales should they be modeled, and what empirical data are needed to parameterize and assess model performance? The primary currencies exchanged across land–ocean–atmosphere–cryosphere boundaries include water, energy, carbon, nutrients (e.g., nitrogen, phosphorous, iron, etc.), and oxygen[91].

From the watershed side, we argue that the model domain in most need of improvement is the low elevation shoreline zone, as modeled hydrological runoff and associated nutrient and carbon loads must pass reactively through marsh and deltaic regions before fluxes can be accurately transmitted to the receiving waterbody; reactive transport through the marsh system is closely linked to variations in water level[92]. A study utilizing such reactive transport models in the southeastern US concluded that small increases in water level due to sea-level rise may increase nutrient export in marshes that have elevations near mean high water, but the opposite effect will occur in marshes with lower elevations[92]. Incorporating these processes in the land components of ESMs will allow improved but one-way computation of reactive transport through the marshes to the receiving water models. This would be a significant improvement over their current functionality, in which estuarine and coastal processes including fluxes of nutrients and particulate and dissolved OC[2,27], and gradient-driven baroclinic exchange between the estuaries and the ocean[93] are incorrectly represented, without sufficient resolution to resolve these processes or the net sinks of carbon and nutrients in estuaries.

When the challenge is evaluated from the ESM ocean components, improving coastal interface representation takes on a larger geographical and biogeochemical meaning. Continental coastlines in ESMs are typically represented by large grid cells; a single cell may encompass an entire estuary. As a result, sediment, carbon, and water delivered to the cells are fully mixed and diluted by the cell size and cannot accommodate complex biogeochemical interactions that occur in tidal rivers, estuaries, and the continental shelf. The central problem, in this as in a number of other ESM modeling domains, is how to model grid-averaged fluxes that may critically depend on subgrid-scale heterogeneities[94]. Some global climate models have approached this issue by using estuarine box model approaches, while regionally-refined or Voronoi meshes (shrinking the size and increasing the number of grid cells in the terrestrial–aquatic interface and other critical regions) are other options[95]. These efforts successfully reconstructed observational data, and should be further used for hind- and fore-casting under specific scenarios defined as pressing needs by the scientific community.

Present state-of-the-art regional scale estuarine models (e.g., FVCOM-ICM, SCHISM, and ROMS) simulate estuarine hydrodynamics and biogeochemical processes in a robust manner[96]. This is particularly true for the hydrodynamic components of these 3D baroclinic tools that use turbulence closure schemes for parameterizing eddy viscosity and mixing processes. As a result, the models accurately reproduce tidal circulation, stratification, and exchange flows in the estuaries extending from the upstream river inflow boundary to the ocean boundary typically set at the continental shelf[54]. When applied in high resolution over the nearshore intertidal regions, the models employ wetting and drying techniques to represent flooding[97] and are able to represent tidal processes over tidal distributaries, tidal flats and marsh regions. In addition, researchers have developed modules for submerged aquatic vegetation and tidal marshes, and sediment diagenesis, allowing explicit implementation of known marshes within the estuaries[98]. For example, one study utilizing FVCOM concluded that restored floodplain wetlands contribute large amounts of organic matter to estuaries, aiding in the restoration of historic trophic structure across the coastal interface[96]. An application of the ROMS model to several Northeastern US estuaries demonstrated that the length scale ratio between tidal excursion and salinity intrusion is one characteristic that can be used to broadly distinguish estuarine hydrodynamic regimes[93]. However, the implementation of the biogeochemistry of tidal marshes and submerged aquatic vegetation in fine-scale 3D coastal and estuarine models is an area of emerging technology and requires dedicated research efforts. One aspect missing from these estuarine models is groundwater–surface water interactions and intrusion of seawater into aquifers; this predictive capability has been developed as a distinct class of groundwater models such as SEAWAT[99] and SUTRA[100] though field measurements are still needed to further understanding. Applications of SEAWAT, which does not simulate unsaturated flow (i.e., the water table only rises due to flow through the saturated zone), have shown that the

model performs well when the ocean–aquifer interface is steep but performs poorly when the slope decreases[99].

Representing disturbance and hot spots/moments in the above model framework adds additional complexity. While current ESMs are designed to capture and model forcings such as regional weather and sea-level rise[90], resulting disturbances e.g., coastal flooding, are not represented, although finer-scale models are capable of accurately predicting storm surge and flooding over complex landscapes[101]. Future ESM refinements of disturbance-representation should thus focus on shifting/compound drivers of coastal ecosystem function (e.g., surface vegetation response to flooding) and hydrology (e.g., groundwater inundation versus riverine or tidal flooding), interactivity of biogeochemical cycling and elemental stores with all ESM components (e.g., redistribution of SOC due to coastal erosion), and inclusion of other types of disturbances (e.g., low-tide rainfall, permafrost thaw). Such disturbance regimes have been identified as important components of local to regional scale response of coastal systems to change[15,81].

## Recommendations

**Bridging the gap between model scales**. We recommend three potential approaches for improving coastal interface representation in ESMs with varying levels of process-level detail. The first approach is a simplified representation that involves finding generalizable features of coastal ecosystems that can be binned as different coastal interface functional types (Fig. 5a). These functional types could include distinct tidal river classifications based on topographic regimes (i.e., passive and active margins) and stream order (Fig. 2b), estuarine regimes (e.g., salt wedge, fjord, well-mixed; Fig. 2c), intertidal ecosystems (e.g., tidal flats, deltas, saltmarshes; Fig. 2d), and shoreline ecosystems (e.g., rocky, sandy, coastal forest). On the ocean side of the interface, this could involve analytical solutions based on bulk properties such as mean estuarine water column depth, flow/depth-averaged salinity gradients, and mixing characteristics (eddy viscosity) to parameterize exchange flows, flushing, and loading. This approach provides a practical simplification that would allow an improvement over the present coastal interface representation in ESMs. Instead of only classifying a pixel as some fraction land and some fraction ocean, a portion of the pixel would also be classified as a certain type of tidal river, estuary, intertidal, and shoreline ecosystem, which is a significant improvement over the current state of the art. However, this would not allow dynamic two-way coupling of land, atmosphere, and ocean models. It is also difficult to incorporate how the hot spot and hot moment dynamics described previously would be treated in such a framework, except perhaps as long-term averages.

On the land side, column-based models that represent changes in vegetation and marsh biogeochemistry would build off existing ESM components. The models could be used to assess carbon stores and losses, and simulate complex biogeochemical cycles in response to simplified hydrological forcings related to sea-level rise and salinity changes. These model structures may have limitations in capturing lateral fluxes between columns, groundwater–surface water interactions, and geomorphic change. As a result, again, realistic representation of hot spots/moments would be limited or nonexistent.

The second approach is a detailed, brute force 3D representation of coastal systems around the world (Fig. 5b). All major coastal seas, estuaries, and deltas worldwide would be explicitly simulated through nesting or similar two-way coupling procedures. The estuarine models with tidal marshes would provide a complete representation of coastal interface processes that allows feedback between each component and provide the most robust

representation of hot spots/moments and disturbance effects. This approach requires the development of estuarine circulation and biogeochemistry models of all major estuaries worldwide. In many developed areas, such models have already been developed[102,103] and can serve as the starting point. In remote regions, model development may be performed using climatological information, but in these cases in situ data for model calibration/validation may be limited or unavailable.

Major challenges to such a process-rich modeling approach include the coupling of model domains (atmosphere, land, ocean, surface, and subsurface) at appropriate scales, the computational resources to simulate these systems at resolutions needed to capture the process dynamics and feedbacks that distinguish individual regions from others, and the large (and perhaps impractical) efforts required for model development and, crucially, maintenance and accessibility to a range of users. On the plus side, however, the resolution demand includes both temporal and spatial scales needed to accurately represent both hot spots and hot moments. Fundamental research is still needed to understand these scales and whether the integrated, both in time and space, impact of hot spots and hot moments justifies the computational costs of explicitly representing them.

The third approach is a combination of simplified and detailed representation of the world's coastline, whereby existing high-resolution estuarine, land and ecogeomorphic, and integrated hydrological models are used to leverage community efforts as virtual field sites for developing reduced-order modeling approaches for existing ESM components. For example, physical Earth system modeling parameterization in the land, river and ocean components of the ESM could be employed at spatially-variable resolutions near the coasts to allow process-rich fidelity to span the scale between the largest ESM scales (100 km) and smallest estuarine and marsh scales (1 m).

We suggest that the brute force approach on its own is unrealistic and undesirable; it is also inconsistent with the central goals of ESMs, which center on abstracting and thus understanding the complete Earth system climate. Thus, melding both approaches is needed to leverage existing ESM capabilities present in land, river, and ocean modeling to enable them to predict under-resolved physics with enhanced fidelity, leveraging the information already available in existing site implementations of estuarine models. Under this framework, the land and atmosphere components of the ESM should provide worldwide watershed loading (flow and nutrients) and weather forcing (long and shortwave solar radiation and wind forcing). While processes such as worldwide watershed loading, weather forcing, and coastal flooding are actively being developed into ESM frameworks[104], their full incorporation into coupled ESMs is necessary before coastal interface representation is possible to address. Improving existing coarse resolution shoreline pixels with explicit 3D model representations using coupled high-resolution components requires, at a minimum, a synthesis of existing observational data at coastal interfaces that could leverage such incorporation. Such efforts could also be combined with spatially-distributed and/or grouping-based sensitivity analyses to further identify a reduced number of the most robust parameters to incorporate into ESMs[105]. Box 1 outlines the recommended criteria for process/element representation within the framework of system classifications to embark on such approaches for coastal interfaces. While representing the coast will consume additional computing resources, we suggest that this will have a low overall burden considering the small global extent of the coastline and the relatively low computational cost of existing land models (relative to the ocean and atmosphere). We posit that the outsized role of coastal systems on global biogeochemical cycles merit any additional computing resource needs. For example, the ocean

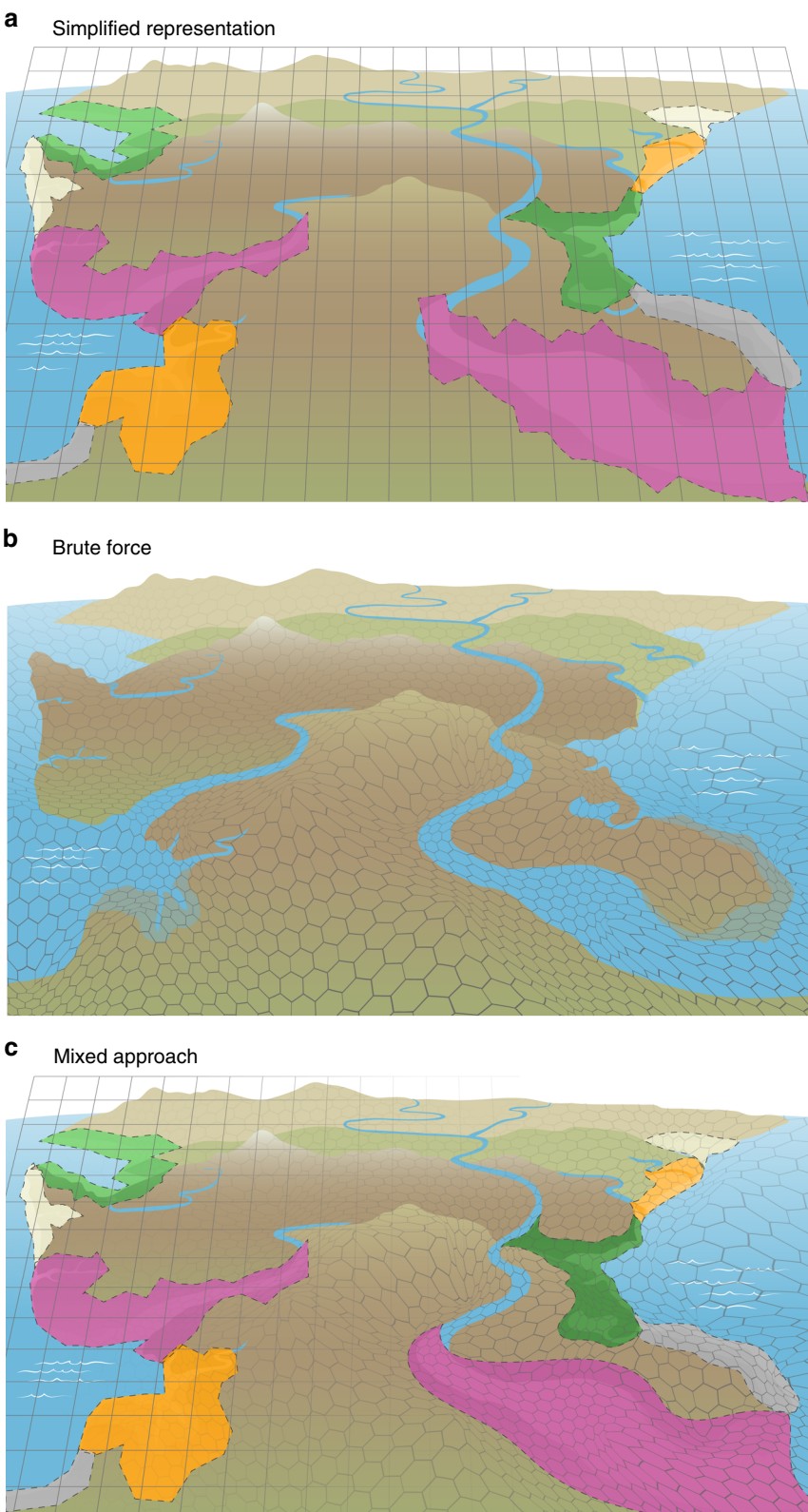

**Fig. 5 Representing coastal interfaces in Earth system models. a** Perhaps the simplest approach would be to classify coastal interfaces based on a series of functional types for their main features (i.e., different types of tidal rivers, estuaries, intertidal ecosystems, and shoreline ecosystems; Fig. 2). Process parameterizations derived from synthesized data would be applied to the fraction of a pixel occupied by each feature rather than the current state of the art, which assigns some fraction of coastal pixels as land and some fraction as ocean. **b** The most sophisticated approach would be to couple high-resolution regional coastal interface models with coarser resolution Earth system models using a variable pixel size (i.e., Voronoi mesh). **c** Perhaps the most feasible and robust approach would be a combination of the two, whereby existing or strategically developed high-resolution models are coupled, and classifications of functional types are applied to systems where data required for high-resolution models are not available.

**Box 1. | Key attributes and processes in the coastal interface that should be represented in ESMs either empirically (i.e., parameterized) or mechanistically (i.e., process)**

| Processes and attributes | Global impact | Relevant stress and disturbance | References |
|---|---|---|---|
| Greenhouse gas fluxes from tidal rivers, nearshore estuarine systems, and marshes | Poorly quantified for coastal systems. Tidal river fluxes not included in global budgets, but may contribute significantly | SLR, salinization, extreme events, temperature, land use change | 8–10,22,35,46,59,118 |
| Carbon sequestration in coastal ecosystems | Equal to 10% net residual land and 50% net marine sediment sinks | SLR, salinization, land use change, temperature | 9,26,36,37,48,50,51,53,55,56,75,83,85 |
| Nutrient and organic carbon cycling | Coastal interface acts as a source and/or sink of biogeochemically important elements (e.g., nutrients) that influence productivity of coastal and marine systems | SLR, water level fluctuations, hypoxia, anthropogenic structures/activities | 2,7,19,25,27,39,43,54,58,74,87,92,98 |
| Hydrodynamics | Controls timing and magnitude of material storage, processing and export | SLR, water level and river discharge fluctuations, storms, anthropogenic structures/activities | 20,21,23,28,92,95,100 |
| Gradients in vegetation communities | Influences biogeochemical functions described above and interacts with geomorphological processes | SLR, salinization, extreme events, temperature, land use change | 15,30,31,33,52,84,86,126,127 |
| Geomorphology | Controls topography and bathymetry, ultimately influencing vegetation and biogeochemical gradients | SLR, storms, water level and river discharge fluctuations, anthropogenic structures/activities | 6,15,18,26,27,38,77–82,107 |
| Erosion | Increases export of organic and inorganic materials across interface; redeposition also occurs. Net result is dependent on interplay between erosion and landward migration in marshes | SLR, storms, water level and river discharge fluctuations, anthropogenic structures/activities | 26,27,38,77–82 |

and atmosphere modules of the Energy Exascale Earth System Model consume ~90% of the model's computing resources[90]; only a fraction of the 10% used by the land module would be needed for coastal representation.

**Observational and experimental needs**. Achieving the goals described above require mechanistic and quantitative knowledge detailed enough to capture the richness of coastal interface processes, but classified at a broad enough scale to tractably incorporate into modeling frameworks. Therefore, it is crucial to synthesize observations of geographical, geological, geomorphological, biogeochemical, and hydrodynamic conditions and processes across coastal systems from current and future efforts (Box 1). Local geological formations, climate conditions, plant functional type distribution, and hydrology/hydrodynamics, that all interact to form terrestrial soils and sediments, determine the physical topography/bathymetry of coastal interfaces. The synthesis of commonalities and differences among coastal system types will allow for the assessment of continental-scale gradients and trends[106]. For example, shallow water depositional settings can be generalized based on the relative influence of river discharge and tidal versus wave-induced erosion[107] along with water mixing and upwelling regimes[23] (Fig. 2). Investigations of the spatial and temporal extent of the intertidal zone in relationship with rainfall intensity and frequency and the context of changing climate and local/regional sea-level rise is also an area that should be explored to evaluate the role of processes such as low-tide rainfall[62]. These types of categorizations and parameterizations of coastal interface features may prove a useful means for managing their complexity in ESMs.

As another example, continental-scale gradients in coastal ecosystem types can be elucidated through the classification of coastline type, dominant flow regimes and tidal forcings, and climatological regions[23] (Fig. 2). Large scale environmental network programs exist that inherently encompass such heterogeneity of observations in coastal ecosystems (e.g., in the USA, the Long Term Ecological Research Network (LTER), the National Ecological Observatory Network (NEON), the National Oceanic and Atmospheric Administration's Sentinel Site Program, and the Coastal Carbon Research Coordination Network), and such efforts should be further leveraged and expanded to address data needs for model incorporations as many areas of the world remain unmonitored[106,108]. Efforts are needed to increase synergy among existing observational data streams to cohesively span larger spatiotemporal scales, allowing for predictive understanding[108]. Synthesis of observational data is one way to address continental-scale patterns and differences among ecoregions[109], and can further identify controlling factors to include into ESM process representation[110]. If appropriately synthesized, continental-scale networks can address both the need for broad classifications of coastal interface features (Fig. 2) and the intensive finer-scale observations and experimentation needed to inform fundamental understanding of hot spots/moments.

Remote sensing is another way to couple and scale from individual measurements to regional and global predictions, and can be particularly powerful when coupled to large-scale experimental and/or observational efforts[111]. While applications of multispectral and hyperspectral remote sensing in coastal settings are challenging due to high optical complexity[112], they can provide extended observations of biogeochemical processes in coastal interfaces over seasons to years[113], characterizing turbidity[114] and a variety of water quality parameters such as suspended sediments[115,116], supporting estimates of sediment export to oceans[117]. These measurements can also be used to estimate dissolved organic carbon concentrations, and linked parameters such as $CO_2$ partial pressure[118] in open water surfaces. In addition, remotely sensed data can be used to map

the spatial distribution of plant functional types and aboveground biomass[119,120]. On the other hand, active remote sensing from lidar or radar interferometric instruments, combined with field measurements, provide large scale characterization of vertical structure in marine habitats, for example enabling global estimates of height and biomass of mangrove forests[120]. Active remote sensing also provides an efficient tool to study hydrodynamics in coastal settings. Repeat-pass interferometry has been used to measure relative water level changes in wetlands[121], and spaceborne altimetry can measure global variations in sea-level along the coasts[122]. The NASA-ISRO Synthetic Aperture Radar (NISAR) and Surface Water and Ocean Topography (SWOT) missions are planned for launch in early 2022 and the Fall 2021, respectively, and promise valuable measurements of coastal hydrodynamic processes. NISAR will measure relative water level change in coastal wetlands while SWOT will measure global water surface elevation and slope along the coasts, within rivers, estuaries, and deltas, which may resolve relevant coastal hydrodynamic processes[123]. However, to resolve tidal and episodic hydrodynamics processes (e.g., storm surge) along the coasts will require extensive in situ measurement networks[106] and airborne measurements[124].

In addition to and coupled with observations, field and lab-scale experimentation is also essential to developing and validating models[111,125], particularly in coastal interfaces where multiple drivers interact to control ecosystem processes. For example, a 30-year experiment has shown that atmospheric $CO_2$-enrichment enhanced primary productivity for the brackish marsh plant, *Schoenoplectus americanus*, but the allocation of biomass changed to favor smaller/denser stems and expanded roots and rhizomes to alleviate nitrogen limitation[126]. A similar 2-year experiment with the seagrass, *Zostera marina*, showed that enhanced marine $CO_2$ levels (i.e., acidification) yielded significantly higher primary productivity and also greater tolerance to high summer temperatures, suggesting that seagrass could act as a negative feedback for increasing atmospheric $CO_2$ levels, in contrast to calcareous aquatic organisms that will suffer from acidification[127].

The tradeoffs between precision and realism inform experimental design such that the optimal scale and treatments will vary with the particular question; smaller-scale experiments offer higher precision in treatment application and assessment of response variables[128], while larger-scale experiments offer greater realism on ecosystem-scale impacts of treatments such as warming, elevated $CO_2$, salinization, and flooding[16,129]. While it is a great advantage to be able to attribute responses to one global change driver, some experimental treatments, such as warming or flooding, confound proximal drivers, such as soil moisture, salinity, pH, and redox status, that are known to control ecosystem processes and mediate the influence of global change drivers. For instance, a coastal ecosystem warming study will alter temperature but also soil moisture and potentially salinity, such that the influence of each proximal driver is difficult but not impossible to determine[130]. Similarly, a flooding manipulation[16] will simultaneously alter redox status, soil moisture, and salinity. Biogeochemical models handle these variables individually. Especially in coastal interfaces, we need field experiments capable of isolating the effects of proximal drivers to allow for the development and validation of models.

## Concluding remarks

Ecological and biogeochemical processes occurring along coastal interfaces are poorly understood on a mechanistic level and critically underrepresented in current ESMs, impeding our ability to make informed resource management decisions. Because coastal ecosystems are characterized by transport-dependent processes and biogeochemistry operating at fine spatial scales, they are extremely challenging to model with any accuracy or precision. Perhaps the most critical questions to address in future research are: what factors and mechanisms lead to resistance and resilience, or lack thereof, in coastal ecosystems in response to external drivers, including both press and pulse disturbances and what are the reciprocal effects of terrestrial on aquatic ecosystems near the coast and vice versa? Answering them requires suitably identifying the dynamic seaward and landward extent of coastal interfaces; measurements that directly improve on modeling of the coastal interface domain and changes due to external drivers, including two-way exchanges/transformations of carbon, water, energy, nutrients, and sediments; and identifying and quantifying the interactions between hydrodynamics, geomorphology, plants, and microbes in forming coastal ecosystem structure and regulating fluxes/transformations of mass and energy within and across three-dimensional boundaries.

We argue that a predictive understanding of the role of coastal interfaces on a global scale is not a task that can be achieved by any one agency, institution, or researcher, but requires collaboration across scales, disciplines, cultures, and funding agencies. The central mission of improving the representation of coastal interfaces in ESMs can be achieved by combining efforts at existing networks with new networks that address critical gaps described in this review. For example, many research networks focus on addressing site-specific objectives and models rather than an overarching mission across scales. As such, we advocate for widespread collaboration, stronger interoperability, and synergistic investments to address this grand challenge. Tangible first steps include workshops that bring together scientists across disciplines to reach a consensus on areas of need such as those described in this review, followed by more focused efforts to identify synergies between existing observational networks, long-term experiments, and regional models. Finally, the scientific community must identify the strengths of specific funding agencies with respect to the coastal domain and propose joint efforts to effectively leverage these strengths to facilitate representation of coastal interfaces in next-generation Earth system models.

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

## Acknowledgements

Funding for this work was provided by Pacific Northwest National Laboratory (PNNL) Laboratory Directed Research & Development (LDRD) as part of the Predicting Ecosystem Resilience through Multiscale Integrative Science (PREMIS) Initiative. PNNL is operated by Battelle for the U.S. Department of Energy under Contract DE-AC05-76RL01830. Additional support to J.P.M. was provided by the NSF-LTREB program (DEB-0950080, DEB-1457100, DEB-1557009), DOE-TES Program (DE-SC0008339), and the Smithsonian Institution. This manuscript was motivated by discussions held by co-authors during a three-day workshop at PNNL in Richland, WA: *The System for Terrestrial Aquatic Research (STAR) Workshop: Terrestrial-Aquatic Research in Coastal Systems*. The authors thank PNNL artist Nathan Johnson for preparing the figures in this manuscript and Terry Clark, Dr. Charlette Geffen, and Dr. Nancy Hess for their aid in organizing the STAR workshop. The authors thank all workshop participants not listed as authors for their valuable insight: Lihini Aluwihare (contributed to biogeochemistry discussions and development of concept for Fig. 3), Gautam Bisht (contributed to modeling discussion), Emmett Duffy (contributed to observational network discussions), Yilin Fang (contributed to modeling discussion), Jeremy Jones (contributed to biogeochemistry discussions), Roser Matamala (contributed to biogeochemistry discussions), James Morris (contributed to biogeochemistry discussions), Robert Twilley (contributed to biogeochemistry discussions), and Jesse Vance (contributed to observational network discussions). A full report on the workshop discussions can be found at https://www.pnnl.gov/publications/star-workshop-terrestrial-aquatic-research-coastal-systems.

## Author contributions

J.P.M. and N.D.W. co-chaired the workshop and guided the overall motivation and conceptual framework for the manuscript. The Ecosystem-Scale Interactions section was led by E.A.C., H.D., and R.M.P. The Biogeochemical Interactions and Cycles section was led by V.L.B., M.A.G., C.S.H., A.S., P.B.W., and L.W.-M. The Hot Spots and Hot Moments section was led by D.B., E.B.G., R.B.N., and R.V. The Disturbances and Stress at the Coastal Interface section was led by N.K.G., N.G.M., and C.L.O. The Modeling the Coastal Interface and Bridging the Gaps Between Model Scales sections were led by B.B.-L., T.K., J.R., and P.E.T. The Observational and Experimental Needs section was led by A.N.M.-P., M.S., J.A.L., and M.T. Figure 1 was inspired by artwork from P.E.T. Figure 2 was inspired by concepts developed by P.E.T., J.P.M., and C.S.H. Figure 3 was inspired by concepts developed by E.A.C., M.A.G., A.S., M.T., E.B.G., and P.B.W. Figure 4 was inspired by concepts developed by N.G.M. Figure 5 was inspired by concepts developed by T.K., P.E.T., B.B.-L., and N.D.W. Box 1 was developed by A.N.M.-P. Preparation of the manuscript was led by N.D.W. All authors critically reviewed the manuscript in its entirety and approval of its submission.

## Competing interests

The authors declare no competing interests.
