## [Peer Review File · Nature Communications]

Reviewers' comments:

Reviewer #1 (Remarks to the Author):

Ward et al. provide an extensive review of processes within the coastal interface that play a role in biogeochemical cycling and ecosystem dynamics. The authors summarize with 3 recommendations broadly concerning how these processes could be represented in Earth System Models.

The authors do an excellent job of compiling and summarizing the literature concerning the many dynamic processes that make the coastal interface such a complex environment. The manuscript is also very well written, though I have a few comments further below regarding the clarity and purpose of a couple of the figures. I see a lot of value in this manuscript as a review of the coastal environment, yet I'm not sure the manuscript sufficiently addresses the underlying goal described in the title – essentially how this environment should be represented within a global-scale Earth System Model. The crux of the problem is very well described in lines 350-353 of the manuscript, but I did not come away with a great understanding of how the authors propose to actually accomplish this extremely challenging goal. The three recommendations listed are presented very broadly, and the authors seem to essentially reject their 2nd recommendation. Furthermore, the first recommendation is essentially the strategy already employed by the large modeling centers. Recommendation 3 is essentially a compromise, but to my understanding would still require a substantial increase in processes and resolution required in the coastal environment. This translates to an increase in required computational resources, which may be a difficult sell to other ESM component groups (e.g. atmosphere, land, ocean) that want an increase in spatial resolution of the global grid to help resolve model deficiencies. While locally important, I did not think the authors made the case that the ecological hot moments were critical at larger spatial-temporal scales. Yet, the way that each recommendation specifically addressed the capability to represent these hot moments seemed to imply that they were one of the most critical processes to capture.

My suggestion is that the recommendations should be structured more around defining a couple key coastal processes that are currently not represented in Earth System Models and have the greatest effect on large-scale physical, biogeochemical, and ecosystem properties. The authors present some good candidates earlier in the text, including a number of processes that account for biogeochemical signals of a similar magnitude to global sources/sinks. Without this, the overall impression I'm left with after reading the review is that the coastal environment is far too complex to be represented in ESMs, and that we're a long way off from gaining traction on this issue. While I do agree that the problem is immense, I think the authors can provide a couple concrete recommendations for immediate processes that can be represented.

I've listed a few additional comments addressing specific sections below:

Figure 2: I had a hard time understanding what the main takeaways for the top portion of this figure are. There are a number of differences between the active and passive margin side, but without labels I'm not understanding what exactly they are and which are important for the purpose of this review manuscript. Some additional labels may really help clarify this.

Figure 4: I understand the bottom plot, but I'm not following the top plot. The caption says it's not intended to be quantitative, but am I still correct in interpretation that sea level rise will be a greater coastal disturbance than temperature and acidification (due to the larger area under the curve)? What happens when the combined temperature, acidification, and sea level rise curve intersects with the extreme events line? Lastly, the caption doesn't include ocean acidification in the description.

Lines 117-120: I'd recommend re-writing this sentence as it is a bit confusing to parse through.

Line 476: Describes the largest ESM scale as 10km, but shouldn't this be closer to 100km? The typical

grid resolution is larger than 10km.

Figure 2B: Typo in river for the y-axis caption.

Reviewer #2 (Remarks to the Author):

This manuscript is an excellent review of the current status and future needs of coastal interface modeling as it pertains to global-scale biogeochemical and ecological modeling. This manuscript contributes to the growing body of interdisciplinary literature that highlights the dynamic nature of the coastal zone. The authors do an excellent job of stepping through the different processes and interactions that occur across the coastal interface and then discuss the challenges of capturing these dynamics. I was torn as to where the modeling section should have been placed in the manuscript, but ultimately decided that the authors have it in the right spot. Having the modeling section right before the recommendations helps to focus the reader on what is coming next. All of the scientific reviews and claims in this manuscript are sound. My biggest "criticism" is that some of the language is awkward and I have attached a Word document with some suggested changes. Overall this manuscript is an excellent contribution to the literature and I greatly enjoyed the opportunity to read it and review it.

Reviewer #3 (Remarks to the Author):

The paper highlights the global significance of terrestrial-aquatic interfaces, or the coastal zone, in terms of biogeochemistry and climate regulation. The perspective is novel, interdisciplinary, and well supported.

In terms of disturbances/stressors (Section 3.2), it seems that eutrophication and hypoxia are notably missing from this review. These stressors radically transform the biogeochemistry of coastal zones and have large spillover effects into coastal ocean and atmosphere that are relevant at global scales.

The paper could be shortened. I believe parts of the manuscript are repetitive or could be more concise.

Specific locations:

The introduction is long, and repetitive to the sections that follow. A more concise introduction would send the signal that it is an overview of topics to be discussed in more depth in the sections that follow.

Line 306-317: This paragraph is not very informative. "Salt marshes can be either sources or sinks of material"... "they contract or expand" ... "an increased export...although some...is re-deposited" – what can we draw from this discussion and what does this have to do with the section title of disturbances?
Line 335 – 341: This paragraph is vague in its meaning ("compounded disturbance thresholds"?) and struck me as repetitive.

Interdisciplinary use of terms

Some terms do not translate across disciplines, and there are several key terms in this manuscript that are inappropriate, from an ecological perspective.

- Disturbance – In ecology, this word refers to a discrete or punctuated event. By definition, gradual environmental changes such as temperature increase, acidification, and sea level rise should not be characterized as disturbances.

- Shifting baselines (Fig. 4) – In ecology, this term refers to human perceptions of ecosystem changes relative to a historic baseline, and particularly to human underestimation of accumulated change in an ecosystem. The offset in Fig. 4B labeled "shifting baseline" is referring to a difference in a short-term average of environmental conditions/events from present day, i.e. a new normal for ecosystem

conditions. This is a very different concept and needs to be labeled appropriately.

Recommendations for figures and tables

Box 1

- These key points are a concise summary. I think this text would be good as an abstract in place of what is currently written.
- In addition to or instead of Box 1, I recommend the following: It would be high impact to have a bulleted list or table of the key biogeochemical transformations of the coastal interface and their contribution to global cycles. Some of these are described in text, but putting them together in a table would add weight to the argument that coastal interfaces merit inclusion into ESMs.

Box 2

- The processes recommended for inclusion into ESMs in Box 2 need more description or justification, both in the Box and in text. How would including coastal or estuarine hydrology / geomorphology / microbial processing change into ESMs model outputs? It would be really helpful to see a worked example of how including coastal transformations of matter and energy changes model results.
- The descriptor of Box 2 as "common coastal interface elements" is not very meaningful. In particular, the word elements is vague. Is there another way to describe these?

Fig. 1

- This figure could be improved to be a representation of the model revision approaches recommended at the end of the paper with a slight modification to include four levels: current ESMs (pictured at top), Simplified (coastal interface model pictured at bottom is already an interpretation of the described approach), Brute force (include a Voroni mesh and more complex estuarine circulation model), and Mixed approach (a variety of scales, model testing, and/or congruence between small scale and ESM models. This might require more creativity to come up with the best schematic).

Fig. 2C

- Fig. 2C is a bit confusing in that the Continental Shelf is in the center of the diagram, surrounded by Delta and Estuary. Also, are Delta and Estuary meant to be directly in the center of the tidal/wave power spectrum, or more to one side? It is difficult to interpret this graphic.

Minor comments

Line 207: The influence of low-tide rainfall – so interesting!

Line 219: What is meant by "water connectivity and history interact"?

Line 247: What is a "depo-center"?

Line 253: Remove "at a finer spatial scale" – extreme precipitation and storm surge can easily affect an entire estuary, so these could occur at small or large spatial scales.

Line 265: Ebullition of methane – this is really neat, great example of a hot moment! The hotspots, hot moments section of this manuscript is one of the most important insights.

Line 326-334: soil carbon gain due to expansion of mangroves is compared to soil carbon loss due to salinization and conversion of freshwater wetlands to open water in this paragraph. Which of these processes is forecasted to affect a larger area? Which will be more important? Also, rather than conversion to open water, in many places, forested freshwater wetland swamps are converting to tidal marsh, which has a very different implication for soil carbon.

Reviewer comments in red, author response in black

Reviewers' comments:

Reviewer #1 (Remarks to the Author):

Ward et al. provide an extensive review of processes within the coastal interface that play a role in biogeochemical cycling and ecosystem dynamics. The authors summarize with 3 recommendations broadly concerning how these processes could be represented in Earth System Models.

The authors do an excellent job of compiling and summarizing the literature concerning the many dynamic processes that make the coastal interface such a complex environment. The manuscript is also very well written, though I have a few comments further below regarding the clarity and purpose of a couple of the figures. I see a lot of value in this manuscript as a review of the coastal environment, yet I'm not sure the manuscript sufficiently addresses the underlying goal described in the title – essentially how this environment should be represented within a global-scale Earth System Model. The crux of the problem is very well described in lines 350-353 of the manuscript, but I did not come away with a great understanding of how the authors propose to actually accomplish this extremely challenging goal. The three recommendations listed are presented very broadly, and the authors seem to essentially reject their 2nd recommendation. Furthermore, the first recommendation is essentially the strategy already employed by the large modeling centers. Recommendation 3 is essentially a compromise, but to my understanding would still require a substantial increase in processes and resolution required in the coastal environment. This translates to an increase in required computational resources, which may be a difficult sell to other ESM component groups (e.g. atmosphere, land, ocean) that want an increase in spatial resolution of the global grid to help resolve model deficiencies.

Response: We thank Reviewer #1 for the thorough and constructive review. Representing coastal ecosystems in global models is indeed a grand challenge, for which we have provided our perspective on current state of the art and paths forward for addressing outstanding issues. In order to add more specificity to our recommendations for representing the coastal interface in global models we have added a new box (Box 1 in revised manuscript) detailing specific processes that are likely the most critical to represent. This box summarizes the processes described throughout sections 2.1 and 2.2 and the disturbances/stressors that may impact them (e.g. section 3.2). Indeed, we highlight these processes and attributes (i.e., carbon and nutrient cycling, geomorphology, vegetation gradients, etc.) in section 2.1 and 2.2 because they are likely the most important aspects of coastal systems to represent in models. This was not made clear in the original manuscript. We have significantly streamlined the Introduction in response to reviewer 3's request to remove redundancy. In doing so, we also updated the final paragraph of the Introduction clearly describing the goal of the paper with the line in bold pointing out that we will discuss functions that should be represented in ESMs:

Lines 140: “We review what is known about the ecological and biogeochemical function of coastal ecosystems **in the context of the attributes and processes that should be represented in ESMs**. We then provide recommended approaches for advancing representation of the coastal

interface in ESMs in order to improve climate predictions and impacts on the world's economically valuable and densely populated coastal zone.”

We reiterate this point at the beginning of sections 2.1 and 2.2:

Line 151: “This section describes the fundamental ecosystem scale attributes and interactions that define the coastal interface and should be represented in ESMs.”

Line 200: “This section describes the fundamental biogeochemical functions of coastal ecosystems that are likely the most critical to represent in ESMs”

In the revised manuscript we have identified specific processes that should be prioritized. However, we do not provide an assessment of how including these processes in ESMs will quantitatively influence ESM performance. This effort (e.g., model sensitivity analysis) is a research endeavor in its own right, and beyond the scope of a review paper. We briefly discuss this point:

Line 563: “Such efforts could also be combined with spatially-distributed and/or grouping-based sensitivity analyses to further identify a reduced number of the most robust parameters to incorporate into ESMs.”

With regards to the point about Recommendation 1, we disagree with the reviewer's assessment that this is the approach most modeling centers are already using. For example, in the case of the Energy Exascale Earth System Model (E3SM), the coastal interface consists of an unstructured “wetland” with an inconsistently overlapping open ocean. A river routing network laid on top of the land model delivers freshwater to the ocean but there is no energy exchange or biogeochemical processes represented at the margins. In essence a coastal pixel is defined as some proportion land and some proportion open ocean. We have added some discussion on this point, along with a new figure (Figure 5) to clarify our proposed approach and how it is an improvement over the current state of the art, i.e., instead of simply separating a coastal pixel into land and ocean, functional types for coastal estuaries, wetlands, etc could be developed, which would play some parameterized biogeochemical role under this approach (e.g., carbon burial in marsh/mangrove settings):

Line 497: “This approach involves finding generalizable features of coastal ecosystems that can be “binned” as different coastal interface functional types (FIG. 5). These functional types could include distinct tidal river classifications based on topographic regimes (i.e., passive and active margins) and stream order (FIG. 2A), estuarine regimes (e.g., salt wedge, fjord, well-mixed; FIG. 2B), intertidal ecosystems (e.g., tidal flats, deltas, saltmarshes; FIG 2C), and shoreline ecosystems (e.g., rocky, sandy, coastal forest). On the ocean side of the interface, this could involve analytical solutions based on bulk properties such as mean estuarine water column depth, flow/depth-averaged salinity gradients, and mixing characteristics (eddy viscosity) to parameterize exchange flows, flushing, and loading. This approach provides a practical simplification that would allow an improvement over the present coastal interface representation in ESMs. Instead of only classifying a pixel as some fraction land and some fraction ocean, a

portion of the pixel would also be classified as a certain type of tidal river, estuary, intertidal, and shoreline ecosystem, which is a significant improvement over the current state of the art.”

We do think that Recommendation 2 is impractical and too far of a leap from the current state of the art (given existing computing, data, and knowledge limitations). Nonetheless, we think it is worth mentioning to set the stage for Recommendation 3, which is indeed a compromise between 1 and 2 that takes advantage of existing high resolution regional models and regions with rich observational datasets, then applies lessons learned in those regions to poorly studied locations. The lessons learned include data training and changes in the architecture of the model that better represents processes and maximize the existing HPC resources.

The ocean and atmosphere modules of ESMs such as E3SM typically consume the vast majority (e.g., 90%) of computational resources compared to the land model. In our opinion, adding resolution to the coastline, which is a small fraction of the Earth’s surface, will not be unmanageable, particularly when deployed with a Voroni mesh (variable grid size) approach as proposed. We have added a statement to the end of Section 5.1 in this regard:

Line 567: “While representing the coast will consume additional computing resources, we suggest that this will have a low overall burden, considering (i) the small global extent of the coastline, (ii) the relatively low computational cost of existing land models (relative to the ocean and atmosphere), and (iii) the outsized role of coastal systems on global biogeochemical cycles merit any additional computing resource needs. For example, the ocean and atmosphere modules of the Energy Exascale Earth System Model consume ~90% of the model’s computing resources; only a small additional fraction of the 10% used by the land module would be needed for coastal representation.”

While locally important, I did not think the authors made the case that the ecological hot moments were critical at larger spatial-temporal scales. Yet, the way that each recommendation specifically addressed the capability to represent these hot moments seemed to imply that they were one of the most critical processes to capture.

Response: A point that we made in the hotspots section (and have reinforced in the revised manuscript) is that hot spots should be considered across different scales. For example, many papers interpret hotspots/moments as fine scale heterogeneity, e.g., measurement of a high methane flux meters or minutes apart from a low measurement. We provide a broader perspective suggesting that fine scale examples such as the above interact with larger scale processes such as rapid rates of carbon sequestration across entire estuaries. For example, we point out that fjords are hot spots for terrestrial carbon burial on a global scale, which has been well established with field measurements (i.e. sediment cores). Other finer scale processes such as methane bubbling (ebullition) are so poorly quantified via field measurements that we do not know how important they may or may not be at larger scales. Well-established processes such as carbon burial hot spots are completely feasible to represent in ESMs, while poorly understood processes require further mechanistic research before they can be accurately represented. We have added the following sentences at the beginning of section 3.1 to make this point more prominent:

Line 271: “We suggest that hot spots can range from fine scales (e.g., cm^3 , m^2) to the scale of entire estuaries ($10\text{-}1000 \text{ km}^2$) and influence local to global scale material budgets depending on the process. It is both feasible and desirable to represent hot spot dynamics in ESMs that 1) play a significant role on global scale biogeochemical cycles and 2) are empirically understood.”

My suggestion is that the recommendations should be structured more around defining a couple key coastal processes that are currently not represented in Earth System Models and have the greatest effect on large-scale physical, biogeochemical, and ecosystem properties. The authors present some good candidates earlier in the text, including a number of processes that account for biogeochemical signals of a similar magnitude to global sources/sinks. Without this, the overall impression I’m left with after reading the review is that the coastal environment is far too complex to be represented in ESMs, and that we’re a long way off from gaining traction on this issue. While I do agree that the problem is immense, I think the authors can provide a couple concrete recommendations for immediate processes that can be represented.

Response: We thank the reviewer for the thoughtful feedback. Please see our previous response above. In brief, we have included Box 1, which summarizes the specific processes that we advocate for including in ESMs. Likewise, we have framed the entire discussion of coastal ecosystem attributes and processes around those that are likely most important for representing in ESMs. Processes such as carbon burial and greenhouse gas cycling are obvious places to start that may have a significant impact on model performance based on empirical global estimates discussed in this manuscript.

I’ve listed a few additional comments addressing specific sections below:

Figure 2: I had a hard time understanding what the main takeaways for the top portion of this figure are. There are a number of differences between the active and passive margin side, but without labels I’m not understanding what exactly they are and which are important for the purpose of this review manuscript. Some additional labels may really help clarify this.

Response: The primary purpose of the top panel is to visualize the diverse types of coastal shoreline features and attributes that are described in more detail in panels A-C. For example, the “tidal river boundary” indicator shows that tidal extent does not go as far inland for the active margin (shown more explicitly in 2A). Likewise, some of the types of estuaries and shorelines defined in 2B and 2C are shown in the top panel. However, we feel that labeling these components in the top figure will detract from its visual appeal and make the figure too busy. In the case of the tidal river boundary markers, we exaggerated the difference between the passive and active margins (i.e., closer to the coast for the active margin). We acknowledge that the concept of passive and active margins was not adequately discussed in the main text to merit showing in a figure. In response we have added some additional content to sections 3.1 regarding the role of passive vs active margins in shaping hot spots for carbon burial and in 5.1 regarding passive vs active margins as one type of ESM classification:

Line 281: “For example, $\sim 18 \text{ Tg C yr}^{-1}$ is buried in fjord sediments, globally, which is equivalent to 11% of marine carbon burial rates; much of this OC is terrestrially-derived owing to the steep topography and a short residence time between terrestrial soils and estuarine sediments in these

environments⁵². This is a feature of landscapes on active margins, whereas lower relief landscapes on passive margins have longer residence times and a greater extent of OC transformation prior to burial (FIG. 2)⁵⁴.”

Line 498: “These functional types could include distinct tidal river classifications based on topographic regimes (i.e., passive and active margins) and stream order (FIG. 2A), estuarine regimes (e.g., salt wedge, fjord, well-mixed; FIG. 2B), intertidal ecosystems (e.g., tidal flats, deltas, saltmarshes; FIG 2C), and shoreline ecosystems (e.g., rocky, sandy, coastal forest).”

Figure 4: I understand the bottom plot, but I’m not following the top plot. The caption says it’s not intended to be quantitative, but am I still correct in interpretation that sea level rise will be a greater coastal disturbance than temperature and acidification (due to the larger area under the curve)? What happens when the combined temperature, acidification, and sea level rise curve intersects with the extreme events line? Lastly, the caption doesn’t include ocean acidification in the description.

Response: We thank the reviewer for the useful feedback. We have adjusted the size of the area underneath each curve to be equal for each disturbance to avoid implying anything about their relative influence. We do not intend to describe the relative magnitude of how each disturbance impacts coastal ecosystems because this is not well-established to our knowledge. The top panel is meant to describe each disturbance individually, thus the intersection between the extreme event line and SLR, etc curve is not intended to mean anything. To clarify this, we changed the panel label to “Individual Coastal Disturbances and Stressors.” The bottom panel is intended to show how these disturbances combine to influence ecosystem vulnerability. To clarify this point we have changed the label to “Ecosystem Vulnerability to Compounding Disturbances.” We have also added “water acidification” to the figure caption. Finally, we added eutrophication and hypoxia to panel A in response to Reviewer 3’s suggestion.

Lines 117-120: I’d recommend re-writing this sentence as it is a bit confusing to parse through.

Response: This sentence has been revised as follows:

Line 166: “In the case of the tidally-influenced reaches of rivers with high discharge such as the Amazon River, the landward salinity intrusion is limited and water can remain fresh some distance offshore onto the continental shelf¹⁰. In contrast, smaller rivers experience significant salinity intrusion within channels and groundwater/soils²⁴.”

Line 476: Describes the largest ESM scale as 10km, but shouldn’t this be closer to 100km? The typical grid resolution is larger than 10km.

Response: That is correct—thanks. We have updated the text with 100km.

Figure 2B: Typo in river for the y-axis caption.

Response: Fixed

Reviewer #2 (Remarks to the Author):

This manuscript is an excellent review of the current status and future needs of coastal interface modeling as it pertains to global-scale biogeochemical and ecological modeling. This manuscript contributes to the growing body of interdisciplinary literature that highlights the dynamic nature of the coastal zone. The authors do an excellent job of stepping through the different processes and interactions that occur across the coastal interface and then discuss the challenges of capturing these dynamics. I was torn as to where the modeling section should have been placed in the manuscript, but ultimately decided that the authors have it in the right spot. Having the modeling section right before the recommendations helps to focus the reader on what is coming next. All of the scientific reviews and claims in this manuscript are sound. My biggest "criticism" is that some of the language is awkward and I have attached a Word document with some suggested changes. Overall this manuscript is an excellent contribution to the literature and I greatly enjoyed the opportunity to read it and review it.

Response: We thank the reviewer for the thorough and constructive review. We have made the changes that you suggested in your marked up version. Responses to the comments are summarized below and all suggestions of minor edits were accepted:

Line 8: By coastal interfaces do you mean these different subsystems combined?

Response: Yes. Coastal interface is used throughout the manuscript to refer to the collection of different coastal ecosystems (e.g., marshes, tidal rivers, estuaries, etc). We have modified the abstract in response to other reviewer comments. The first two sentences, specifically, should clarify what we mean by coastal interface (and note that a word limit of 100 leaves little room for more nuance):

Line 90: "Along coastal interfaces, components of the Earth system interact to regulate ecosystem functions and Earth's climate. Between the land and ocean, diverse coastal ecosystem types transform, store, and transport material."

We also make this point clear in the revised first sentence of the introduction with more nuance:

Line 100: "The coastal interface, where the land and ocean realms meet (e.g., estuaries, tidal wetlands, tidal rivers, continental shelves, and shorelines), is home to some of the most biologically and geochemically active and diverse systems on Earth ¹."

Line 13: These sentences are somewhat redundant. Could be compressed into just one

Response: Agreed, we have changed these two sentences into one as follows:

Line 100: "The coastal interface, where the land and ocean realms meet (e.g., estuaries, tidal wetlands, tidal rivers, continental shelves, and shorelines), is home to some of the most biologically and geochemically active and diverse systems on Earth ¹."

Line 42: Citation for this

Response: We have added the following reference:

Windham-Myers, L., Cai, W.J., Alin, S., Andersson, A., Crosswell, J., Dunton, K.H., Hernandez-Ayon, J.M., Herrmann, M., Hinson, A.L., Hopkinson, C.S. and Howard, J., 2018. Chapter 15: Tidal wetlands and estuaries. *Second State of the Carbon Cycle Report (SOCCR2): A Sustained Assessment Report*, edited by: Cavallaro, N., Shrestha, G., Birdsey, R., Mayes, MA, Najjar, RG, Reed, SC, Romero-Lankao, P., and Zhu, Z., US Global Change Research Program, Washington, DC, USA, pp.596-648.

Line 44: You mention later on how carbon can be exported from these environments via land use change and erosion, may be good to mention it here as well.

Response: We have deleted this sentence from the Introduction in response to reviewer 3's request that we make the Introduction more streamlined. This point is still discussed later on, however.

Line 52: Variability and vulnerability?

Response: Changed as suggested

Line 52: This is great information, but a really long sentence. Suggest breaking it up.

Response: Broke into two sentences as suggested

Line 88: Might want to add a clarifying statement in here about this. This is related to a change in methane and CO₂, correct? I suspect this manuscript will be of interest to readers from a wide range of fields so it would be helpful just to make that clear.

Response: As per our previous comment, we have removed redundancy between the Introduction and main text. This sentence can now be found in section 3.2 and was revised as follows:

Line 360: "Further, the ecological structure of coastal ecosystems is already experiencing the effects of sea-level rise with coastal forest boundaries retreating inland¹⁵ and salinization of tidal freshwater systems shifting their function and related rates of carbon burial and greenhouse gas (i.e., CO₂ and CH₄) emissions⁴⁷."

Line 93: Promote a?

Response: Changed to: "We **advocate for improved** mechanistic understanding of coastal interfaces from ecological and functional perspectives, the impact of coastal interfaces on global biogeochemical cycling and climate, and the effect of disturbances on coastal interfaces across a range of spatiotemporal scales."

Line 112: Ecological/physical or both?

Response: Added both clarifying terms

Line 122: How specifically? Maybe add a sentence in that fleshes this out more?

Response: Added this clarifying sentence:

Line 171: “For example, tidal exchange can both deposit marine-derived material onto terrestrial landscapes²⁵ and export terrigenous material to the sea^{26,27}. “

Line 137: Soil organic may be better here

Response: Agreed, and changed accordingly.

Line 162: There are some though. Thinking about Kirwan and Mudd’s work as well as James Morris’ work.

Response: We added the following reference to this sentence:

Kirwan, M.L., Guntenspergen, G.R., d’Alpaos, A., Morris, J.T., Mudd, S.M. and Temmerman, S., 2010. Limits on the adaptability of coastal marshes to rising sea level. *Geophysical research letters*, 37(23).

Line 176: Time and space? Or just space here?

Response: The cited reference refers to spatial scales, as well as the discussion that follows, so we have added “spatial” as a clarifier.

Line 192: I would suggest making this a new paragraph.

Response: Agreed and changed accordingly.

Line 219: Microbial activity?

Response: Changed to:

Line 249: “At the pore-scale microbial activity, hydrologic connectivity, and drought legacy interact to regulate ecosystem functions⁴³.”

Line 254: Fluctuations of what? Physical, biogeochemical, and ecological processes?

Response: Changed as follows and also split into 2 sentences:

Line 293: “Coastal ecosystems are sensitive to rapid and disproportionate hydrological and biogeochemical fluctuations with terrestrial, atmospheric, and oceanic origins including extreme precipitation events⁵⁶, snow/ice melt¹⁰, accumulation and enhanced dry deposition of atmospheric pollutants⁵⁷, extreme high tides, and storm surges^{45,58}. Thus, hot moments—short

time periods with disproportionately high metabolic rates—may play a prominent, but typically ignored role in coastal ecosystem biogeochemical cycling. “

Line 315: Could also cite Theuerkauf et al. 2015 here.

Response: Agreed--reference added

Line 323: Theuerkauf and Rodriguez 2017 – Barrier island transgression and carbon- may be a better citation here. Also the work of Lorenzo-Trueba and Ashton.

Response: Agreed. Theuerkauf and Rodriguez 2017 was added, however we are up against the reference limit so did not add the other suggestions.

Line 417: This paragraph seems a bit out of place. I would suggest either placing it into more context with the rest of the manuscript, or incorporating the content here in other paragraphs.

Response: We have deleted this paragraph considering atmospheric deposition is mentioned earlier in the hot moments section. Likewise, in the recommendations section we mention that a challenge is coupling different model domains (atmospheric, land, ocean) at relevant scales.

Line 430: Such as? This is clear when you look at the figure, but perhaps a couple of examples would help to make it clear as you are reading.

Response: We have changed these sentences as follows and also added a new figure to visualize the 3 recommended modeling approaches:

Line 497: “This approach involves finding generalizable features of coastal ecosystems that can be “binned” as different coastal interface functional types (FIG. 5). These functional types could include distinct tidal river classifications based on topographic regimes (i.e., passive and active margins) and stream order (FIG. 2A), estuarine regimes (e.g., salt wedge, fjord, well-mixed; FIG. 2B), intertidal ecosystems (e.g., tidal flats, deltas, saltmarshes; FIG 2C), and shoreline ecosystems (e.g., rocky, sandy, coastal forest). “

Line 526: Synthesized?

Response: Changed

Line 527: Broad classifications of what?

Added the following to tie this sentence to the “simplified classification” recommendation in section 5.1:

Line 606: “If appropriately synthesized, continental scale networks can address both the need for broad classifications of coastal interface features (FIG 2) and the intensive finer scale observations and experimentation needed to inform fundamental understanding of hot spots/moments.”

Line 549: Wetlands?

Response: Changed

Line 551: Which may- rather than promising?

Response: Changed

Line 554: Field or lab or both?

Response: Added “field and lab-scale experimentation”

Line 576: This is the first time this acronym is used. I would suggest either defining it in the introduction or not using it here.

Response: We thank the reviewer for the comment. This was left in from an old draft that used different nomenclature. We have replaced “TAI ecosystems” with “coastal ecosystems” for consistency.

Line 595: Perhaps put forth a suggestion for how specifically to do this. This is a great and big goal so an example of how to do this would be helpful for making it clear that this is possible (which I think it is!).

Response: We thank the reviewer for their positive comments and optimism! We have added the following text to the end of the paper with a proposed strategy for accomplishing the goals outlined in the paper:

Line 683: “Tangible first steps include workshops that bring together scientists across disciplines to reach a consensus on areas of need such as those described in this review, followed by more focused efforts to identify synergies between existing observational networks, long term experiments, and regional models. Finally, the scientific community must identify the strengths of specific funding agencies with respect to the coastal domain and propose joint efforts to effectively leverage these strengths to facilitate representation of coastal interfaces in next generation Earth system models.”

In fact, the first step we describe in the revised manuscript is an action that the authors of this paper have already initiated. This review paper is the product of a workshop held one year ago aimed at reaching a consensus on how to begin meaningfully representing coastal systems in ESMs. As described in the updated text, the next step is to work towards bringing together different observational networks, modeling groups, and researchers performing long term experiments to synergize their efforts towards a central mission. This is an activity that we are actively pursuing, and our hope is that this review paper prompts more widespread efforts related to this effort. Finally, we must convince funding agencies that 1) this is an important topic (we hope that this review paper helps) and 2) each agency has a special role to play in this research frontier. The US Department of Energy recently approved funds in their budget for coastal work that may fill some gaps identified in this paper, and we are hopeful that activities such as our

workshop and review paper will keep the momentum for integrated coastal science growing across funding agencies.

Reviewer #3 (Remarks to the Author):

The paper highlights the global significance of terrestrial-aquatic interfaces, or the coastal zone, in terms of biogeochemistry and climate regulation. The perspective is novel, interdisciplinary, and well supported.

In terms of disturbances/stressors (Section 3.2), it seems that eutrophication and hypoxia are notably missing from this review. These stressors radically transform the biogeochemistry of coastal zones and have large spillover effects into coastal ocean and atmosphere that are relevant at global scales.

Response: We thank the reviewer for their thoughtful comments. Indeed eutrophication and hypoxia are critical to consider with respect to the resilience and biogeochemical function of coastal ecosystems. We have included both eutrophication and hypoxia in a revised figure 4 (see more detail in response to comments on figure 4 below) and have added the following text to a revised section 3.2:

Line 367: “ Eutrophication of estuarine waters occurs as a result of both natural episodic nutrient inputs and long-term changes in land use practices (e.g., agriculture, septic systems, nitrogen fixing vegetation, etc.), and in some cases results in hypoxic conditions that can harm fish and wildlife; hypoxia occurs due to both natural and anthropogenic causes ⁷³.“

The paper could be shortened. I believe parts of the manuscript are repetitive or could be more concise.

Specific locations:

The introduction is long, and repetitive to the sections that follow. A more concise introduction would send the signal that it is an overview of topics to be discussed in more depth in the sections that follow.

Response: We have made major revisions to remove redundancy. First we removed several paragraphs from the introduction and merged this content with the main subsections. For example, the original version of manuscript discussed the biogeochemical role of coastal systems on a global scale. This was integrated with quantitative information on coastal C cycling in section 2.2. Likewise we merged a paragraph on coastal disturbances with section 3.2. We also integrated content in lines 129-138 of the original manuscript (Section 2.1) that very briefly discussed the influence disturbances and stressors on wetland structure to section 3.2 to avoid redundancy.

Line 306-317: This paragraph is not very informative. “Salt marshes can be either sources or sinks of material”... “they contract or expand” ... “an increased export...although some...is re-deposited” – what can we draw from this discussion and what does this have to do with the section title of disturbances?

Response: As part of our above effort to reduce redundancy and increase conciseness, we have deleted many aspects of this paragraph that were not informative. The paragraph has been updated as follows:

Line 376: “Shifts in the interaction between freshwater hydrology and tidal influences due to sea-level rise, delta subsidence or anthropogenic changes (e.g., impoundments) will impact coastal interface geomorphology, such as delta evolution, riverine and coastal sedimentation, and wetland ecological/physical structure²⁰. Changes in sediment supply may be considered a stress that alters the evolution of wetland structure and function, although episodic events such as landslides and volcanoes are disturbances under the return-interval based terminology adopted here⁶⁶. Though meta-analyses have shown that salt marshes can keep pace vertically with sea-level rise⁷⁶, the lateral position of marshes is not as stable as they narrow or expand depending on the net sediment budget⁷⁷ and external stressors and disturbances such as waves, storm surge, and sea-level rise. The contraction of marsh area likely produces an increased export of organic and inorganic material across the coastal interface^{26,78} although some portion of material is re-deposited on the marsh plain during the landward transgression process⁷⁹.”

Line 335 – 341: This paragraph is vague in its meaning (“compounded disturbance thresholds”?) and struck me as repetitive.

Response: We have modified the paragraph to emphasize that this is one of the main summary points of the section we’d like to convey; we must be able to predict the net result of multiple disturbances/stressors on an ecosystem. Furthermore, we argue that many of these disturbances and stressors are compounding (e.g. Figure 4).

Interdisciplinary use of terms

Some terms do not translate across disciplines, and there are several key terms in this manuscript that are inappropriate, from an ecological perspective.

- Disturbance – In ecology, this word refers to a discrete or punctuated event. By definition, gradual environmental changes such as temperature increase, acidification, and sea level rise should not be characterized as disturbances.

Response: We thank the reviewer for pointing out this discrepancy in terminology. We have updated this paragraph, figure 4, and later references to “disturbances” in the modeling section with more nuance. We now use the following definitions for disturbance and stress:

Line 339: “Coastal ecosystems are broadly sensitive to disturbances and stress from surrounding watersheds and the ocean that result in anomalous (i.e., non-steady state) responses. Disturbance typically refers to events that temporarily alter ecosystem attributes (e.g., plant productivity, GHG fluxes, etc.) but occur infrequently enough to allow for recovery time during which attributes re-establish a “normal” dynamic equilibrium; in contrast, higher frequency or continuous “stress” events permanently shift the trajectory of an ecosystem attribute⁶⁶. Long-term stress to an ecosystem is also referred to as a press as opposed to a pulse disturbance event, and these can interact producing compounding effects⁶⁷.”

- Shifting baselines (Fig. 4) – In ecology, this term refers to human perceptions of ecosystem changes relative to a historic baseline, and particularly to human underestimation of accumulated change in an ecosystem. The offset in Fig. 4B labeled “shifting baseline” is referring to a difference in a short-term average of environmental conditions/events from present day, i.e. a new normal for ecosystem conditions. This is a very different concept and needs to be labeled appropriately.

Response: We have updated the figure accordingly. “Shifting baseline” was changed to “New Ecosystem State.” Likewise, we added hypoxia and eutrophication to panel A and changed the panel labels to “A. Individual Coastal Disturbances and Stressors” and “B. Ecosystem Vulnerability to Compounding Disturbances/Stressors” in response to the previous terminology comment.

Recommendations for figures and tables

Box 1

- These key points are a concise summary. I think this text would be good as an abstract in place of what is currently written.

Response: We thank the reviewer for the suggestion. We have integrated the old Box 1 content with the abstract

- In addition to or instead of Box 1, I recommend the following: It would be high impact to have a bulleted list or table of the key biogeochemical transformations of the coastal interface and their contribution to global cycles. Some of these are described in text, but putting them together in a table would add weight to the argument that coastal interfaces merit inclusion into ESMs.

Response: We thank the reviewer for the suggestion, and we have modified box one to this point. Box one now includes the major biogeochemical processes and transformations at the coastal interface (e.g. carbon and nutrient cycling), their influence and potential importance in global cycles (e.g. important sinks in coastal vegetated systems and important sources in tidal river reaches), and the disturbances which may alter these processes (e.g., SLR, land use change, etc), and a suggestion of how they might be represented in ESMs (process model vs system representation) .

Box 2

- The processes recommended for inclusion into ESMs in Box 2 need more description or justification, both in the Box and in text. How would including coastal or estuarine hydrology / geomorphology / microbial processing change into ESMs model outputs? It would be really helpful to see a worked example of how including coastal transformations of matter and energy changes model results.

Response: We have updated Box 2 (now Box 1) with more specificity. We have also added clarity throughout the text pointing out that the attributes and processes described in the paper are those which are considered priorities to test in new models. In section 2.2 we describe coastal biogeochemical processes that are impactful on the global scale, but not included in models such as carbon burial in blue carbon habitats and CO₂ emissions from tidal rivers. If properly

represented in ESMs these functions would exert a tangible (and dynamic) influence on modeled material budgets, and ultimately atmospheric greenhouse gas content. However, with respect to the last point on seeing a worked example, this is beyond the scope of a review article. This would be novel research, which we are suggesting is needed. The first step of showing a worked example would be adding a coastal module to an existing ESM, which currently does not exist aside from perhaps in early development phases. The next step would be performing a model sensitivity analysis of how a particular process (e.g., blue carbon burial) influences a specific set of model outputs. This is precisely the type of research we are advocating for. We briefly discuss this point in the revised manuscript:

Line 563: “Such efforts could also be combined with spatially-distributed and/or grouping-based sensitivity analyses to further identify a reduced number of the most robust parameters to incorporate into ESMs.”

- The descriptor of Box 2 as “common coastal interface elements” is not very meaningful. In particular, the word elements is vague. Is there another way to describe these?

Response: This box was merged with the new Box 1, and nomenclature considered carefully in respect to this comment. The new edited box replaces the word ‘elements’ with ‘attributes’ and ‘processes’.

Fig. 1- This figure could be improved to be a representation of the model revision approaches recommended at the end of the paper with a slight modification to include four levels: current ESMs (pictured at top), Simplified (coastal interface model pictured at bottom is already an interpretation of the described approach), Brute force (include a Voroni mesh and more complex estuarine circulation model), and Mixed approach (a variety of scales, model testing, and/or congruence between small scale and ESM models. This might require more creativity to come up with the best schematic).

Response: We have added an additional figure (FIG 5) that shows these different proposed approaches. We did not combine it with Figure 1 since Figure 1 demonstrates a simple point in the introduction (ESMs do not represent the coast), whereas these recommendations come much later in the text. We also added Voroni mesh to the bottom panel of Figure 1 for consistency with our recommendations.

Fig. 2C

- Fig. 2C is a bit confusing in that the Continental Shelf is in the center of the diagram, surrounded by Delta and Estuary. Also, are Delta and Estuary meant to be directly in the center of the tidal/wave power spectrum, or more to one side? It is difficult to interpret this graphic.

Response: This figure is a simplified version of a classic geomorphology figure. We purposely simplified the visual representation of these coastal features to make the point that these features could perhaps be represented in a simple fashion in models (e.g., as a “binned” classification that doesn’t necessarily embody all complex features of a specific region/locality). However, our original caption was overly simple and did not adequately describe the figure. We have updated the caption to describe that the top and bottom represent regressive and transgressive

environments and that continental shelves and tidal flats are features found in both types of environments:

Line 732: “Classifications of shallow water depositional environments along the coast can be categorized based on the ratio of wave power to tidal power and whether they are regressive (i.e., net land gain; top half of diagram) or transgressive (i.e., net land loss; bottom half of diagram) environments. The top half of the diagram shows regressive environments such as deltas and strand plains. The bottom of the diagram shows transgressive environments such as estuaries and barrier lagoons. Open coast tidal flats and shelf environments can be linked to either type of coast with shelf width decreasing during regression (adapted from Steel and Milliken¹¹⁰).”

Minor comments

Line 207: The influence of low-tide rainfall – so interesting!

Response: We agree with the reviewer.

Line 219: What is meant by “water connectivity and history interact”?

Response: We have modified this to clarify as follows:

Line 249: “At the pore-scale microbial activity, hydrologic connectivity, and drought legacy interact to regulate ecosystem functions⁴³.”

Line 247: What is a “depocenter”?

Response: The definition of depocenter is: “an area or site of maximum deposition, or the geographic location of the thickest part of any specific geographic unit in a depositional basin.” We have added “areas of maximum deposition” in parentheses after depocenter on Line 284.

Line 253: Remove “at a finer spatial scale” – extreme precipitation and storm surge can easily affect an entire estuary, so these could occur at small or large spatial scales.

Response: We have made the suggested modification and the sentence now starts with “Coastal ecosystems...”

Line 265: Ebullition of methane – this is really neat, great example of a hot moment! The hotspots, hot moments section of this manuscript is one of the most important insights.

Response: We thank the reviewer for the comment.

Line 326-334: soil carbon gain due to expansion of mangroves is compared to soil carbon loss due to salinization and conversion of freshwater wetlands to open water in this paragraph. Which of these processes is forecasted to affect a larger area? Which will be more important? Also, rather than conversion to open water, in many places, forested freshwater wetland swamps are converting to tidal marsh, which has a very different implication for soil carbon.

Response: We have added the following text to address this comment:

Line 372: “Mangrove forests in tropical regions are losing between 0.16 and 0.39% of land area annually to development, aquaculture, and agriculture⁷⁴. Such coastal land alteration has already released large quantities of soil organic carbon to the atmosphere as CO₂, and an estimated 0.15–1.02 Pg C yr⁻¹ continues to be emitted globally⁷⁵.”

Line 404: “Such landscape-level shifts are dependent on a complex interplay between land use (e.g. extent of coastal development), geomorphic conditions and relative sea level rise. For example, direct salt marsh conversion to open water or tidal flats may have a greater importance than mangrove expansion into salt marsh habitats in low relief areas with high relative sea level rise, while tidal freshwater marshes may either increase or decrease in areal extent under mean or max sea-level rise scenarios⁸⁷. Among the largest uncertainties in projecting the future distribution, structure, and function of coastal interfaces is quantifying tipping points for the collapse of ecosystem structure and function⁸⁸ across the coastal domain that incorporates the combined effects of a myriad of disturbances and stressors with compounding impacts (FIG. 4).”

REVIEWERS' COMMENTS:

Reviewer #1 (Remarks to the Author):

The authors have addressed all of my previous comments and I think the manuscript is greatly improved based on the feedback from the reviewers and the revisions the authors have made. I particularly like what the authors have done with the introduction based on the recommendations from reviewer #3. The revised introduction is now significantly more succinct and impactful, and it really helps set the stage for the rest of the manuscript.

Reviewer #2 (Remarks to the Author):

The authors did an excellent job of addressing all of the reviewer's comments and the revised manuscript is substantially improved. Particularly with respect to clarity and redundancy, but also in terms of addressing technical issues such as the scale of hotspots, biogeochemical processes, and suggested model revision approaches. The figures for this manuscript are also greatly improved and help to make the main points of the paper. The revision to the first paragraph also does this and that structure is carried throughout the manuscript. I suggest that this manuscript is ready for publication and should be officially accepted.

Reviewer #3 (Remarks to the Author):

The authors have done an excellent job on this revision, rising to meet the challenges posed by all of the reviewers. I appreciate the depth to which they revised the manuscript, and I find my own comments on the prior submission suitably resolved.

In particular, the shortened introduction is less redundant and the abstract is now on point. The manuscript is now more focused around informing ESMs, which I think is an excellent direction. The new Figure 5, which diagrams the three recommended modeling strategies is illustrative and clear. The ecological terminology of stress and disturbance is now carefully distinguished throughout.

One minor issue - in text references to Box 2, but I believe Box 2 has been removed and there is now only one Box. Likely a simple error.